# SCARF: SELF-SUPERVISED CONTRASTIVE LEARNING USING RANDOM FEATURE CORRUPTION

**Dara Bahri, Heinrich Jiang, Yi Tay, Donald Metzler**
Google Research
{dbahri,heinrichj,yitay,metzler}@google.com

## ABSTRACT

Self-supervised contrastive representation learning has proved incredibly successful in the vision and natural language domains, enabling state-of-the-art performance with orders of magnitude less labeled data. However, such methods are domain-specific and little has been done to leverage this technique on real-world *tabular* datasets. We propose SCARF, a simple, widely-applicable technique for contrastive learning, where views are formed by corrupting a random subset of features. When applied to pre-train deep neural networks on the 69 real-world, tabular classification datasets from the OpenML-CC18 benchmark, SCARF not only improves classification accuracy in the fully-supervised setting but does so also in the presence of label noise and in the semi-supervised setting where only a fraction of the available training data is labeled. We show that SCARF complements existing strategies and outperforms alternatives like autoencoders. We conduct comprehensive ablations, detailing the importance of a range of factors.

## 1 INTRODUCTION

In many machine learning tasks, unlabeled data is abundant but labeled data is costly to collect, requiring manual human labelers. The goal of self-supervised learning is to leverage large amounts of unlabeled data to learn useful representations for downstream tasks such as classification. Self-supervised learning has proved critical in computer vision (Grill et al., 2020; Misra & Maaten, 2020; He et al., 2020; Tian et al., 2019) and natural language processing (Song et al., 2020; Wang et al., 2019; Raffel et al., 2019). Some recent examples include the following: Chen et al. (2020) showed that training a linear classifier on the representations learned by their proposed method, SimCLR, significantly outperforms previous state-of-art image classifiers and requires 100x fewer labels to do so; Brown et al. (2020) showed through their GPT-3 language model that by pre-training on a large corpus of text, only few labeled examples were required for task-specific fine-tuning for a wide range of tasks.

A common theme of these advances is learning representations that are robust to different views or distortions of the same input; this is often achieved by maximizing the similarity between views of the same input and minimizing those of different inputs via a contrastive loss. However, techniques to generate views or corruptions have thus far been, by and large, domain-specific (e.g. color distortion (Zhang et al., 2016) and cropping (Chen et al., 2020) in vision, and token masking (Song et al., 2020) in NLP). Despite the importance of self-supervised learning, there is surprisingly little work done in finding methods that are applicable across domains and in particular, ones that can be applied to tabular data.

In this paper, we propose SCARF, a simple and versatile contrastive pre-training procedure. We generate a view for a given input by selecting a random subset of its features and replacing them by random draws from the features' respective empirical marginal distributions. Experimentally, we test SCARF on the OpenML-CC18 benchmark (Vanschoren et al., 2013; Bischl et al., 2017; Feurer et al., 2019), a collection of 72 real-world classification datasets. We show that not only does SCARF pre-training improve classification accuracy in the fully-supervised setting but does so also in the presence of label noise and in the semi-supervised setting where only a fraction of the available training data is labeled. Moreover, we show that combining SCARF pre-training with other solutions to these problems further improves them, demonstrating the versatility of SCARF and its ability to

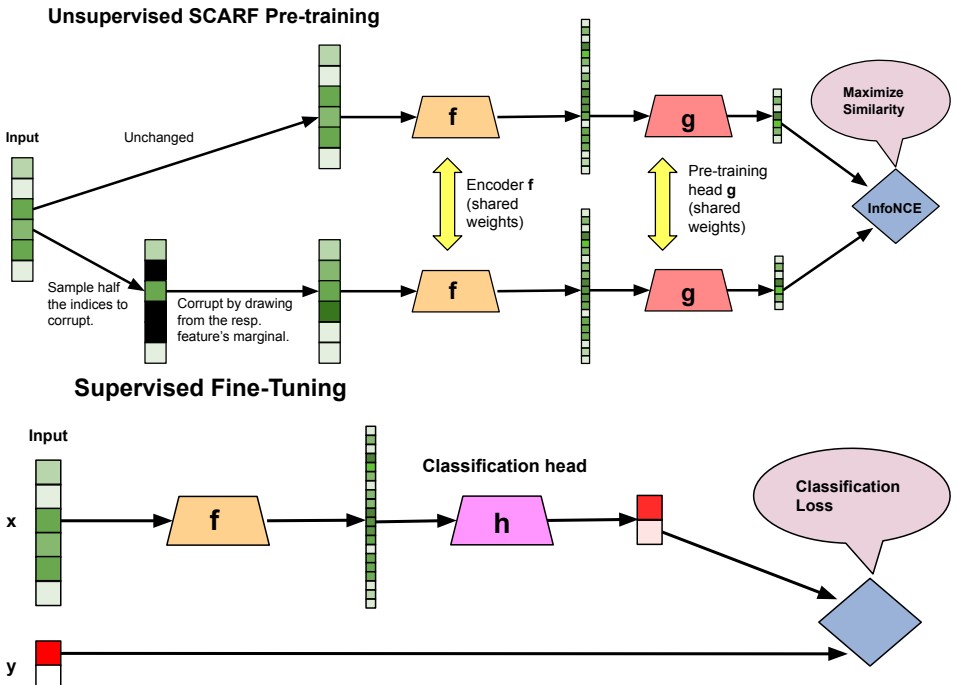

Figure 1: Diagram showing unsupervised SCARF pre-training (**Top**) and subsequent supervised fine-tuning (**Bottom**). During pre-training, networks $f$ and $g$ are learned to produce good representations of the input data. After pre-training, $g$ is discarded and a classification head $h$ is applied on top of the learned $f$ and both $f$ and $h$ are subsequently fine-tuned for classification.

learn effective task-agnostic representations. We then conduct extensive ablation studies, showing the effects of various design choices and stability to hyperparameters. Our ablations show that SCARF's way of constructing views is more effective than alternatives. We show that SCARF is less sensitive to feature scaling and is stable to various hyperparameters such as batch size, corruption rate, and softmax temperature.

## 2 RELATED WORKS

A number of self-supervised learning techniques have been proposed in computer vision (Zhai et al., 2019; Tung et al., 2017; Jing & Tian, 2020). One framework involves learning features based on generated images through various methods, including using a GAN (Goodfellow et al., 2014; Donahue et al., 2016; Radford et al., 2015; Chen et al., 2018), predicting pixels (Larsson et al., 2016), predicting colorizations (Zhang et al., 2016; Larsson et al., 2017), ensuring local and global consistency (Iizuka et al., 2017), and learning synthetic artifacts (Jenni & Favaro, 2018). Most related to our approach are contrastive learning ones (Tian et al., 2019; Hassani & Khasahmadi, 2020; Oord et al., 2018; Henaff, 2020; Li et al., 2016; He et al., 2020; Bojanowski & Joulin, 2017; Wang & Gupta, 2015; Gidaris et al., 2018). In particular, our framework is similar to SimCLR (Chen et al., 2020), which involves generating views of a single image via image-based corruptions like random cropping, color distortion and blurring; however, we generate views that are applicable to tabular data.

Self-supervised learning has had an especially large impact in language modeling (Qiu et al., 2020). One popular approach is masked language modeling, wherein the model is trained to predict input tokens that have been intentionally masked out (Devlin et al., 2018; Raffel et al., 2019; Song et al., 2019) as well as enhancements to this approach (Liu et al., 2019; Dong et al., 2019; Bao et al., 2020; Lample & Conneau, 2019; Joshi et al., 2020) and variations involving permuting the tokens (Yang et al., 2019; Song et al., 2020). Denoising autoencoders have been used by training them to reconstruct the input from a corrupted version (produced by, for example, token masking, deletion, and infilling) (Lewis et al., 2019; Wang et al., 2019; Freitag & Roy, 2018). Contrastive approaches

---

**Algorithm 1** SCARF pre-training algorithm.

---

1: **input:** unlabeled training data $\mathcal{X} \subseteq \mathbb{R}^M$, batch size $N$, temperature $\tau$, corruption rate $c$, encoder network $f$, pre-train head network $g$.
2: let $\widehat{\mathcal{X}_j}$ be the uniform distribution over $\mathcal{X}_j = \{x_j : x \in \mathcal{X}\}$, where $x_j$ denotes the $j$-th coordinate of $x$.
3: let $q = \lfloor c \cdot M \rfloor$ be the number of features to corrupt.
4: **for** sampled mini-batch $\left\{ x^{(i)} \right\}_{i=1}^N \subseteq \mathcal{X}$ **do**
5:     for $i \in [N]$, uniformly sample subset $\mathcal{I}_i$ from $\{1, ..., M\}$ of size $q$ and define $\tilde{x}^{(i)} \in \mathbb{R}^M$ as follows: $\tilde{x}_j^{(i)} = x_j$ if $j \notin \mathcal{I}_i$, otherwise $\tilde{x}_j^{(i)} = v$, where $v \sim \widehat{\mathcal{X}_j}$.     # generate corrupted view.
6:     let $z^{(i)} = g\left( f\left( x^{(i)} \right)\right)$, $\tilde{z}^{(i)} = g\left( f\left( \tilde{x}^{(i)} \right)\right)$, for $i \in [N]$.     # embeddings for views.
7:     let $s_{i,j} = z^{(i)\top} \tilde{z}^{(j)} / \left( \|z^{(i)}\|_2 \cdot \|\tilde{z}^{(j)}\|_2 \right)$, for $i, j \in [N]$.     # pairwise similarity.
8:     define $\mathcal{L}_{\text{cont}} := \frac{1}{N} \sum_{i=1}^N - \log \left( \frac{\exp(s_{i,i}/\tau)}{\frac{1}{N}\sum_{k=1}^N \exp(s_{i,k}/\tau)} \right)$.
9:     update networks $f$ and $g$ to minimize $\mathcal{L}_{\text{cont}}$ using SGD.
10: **end for**
11: **return** encoder network $f$.

---

include randomly replacing words and distinguishing between real and fake phrases (Collobert et al., 2011; Mnih & Kavukcuoglu, 2013), random token replacement (Mikolov et al., 2013; Clark et al., 2020), and adjacent sentences (Joshi et al., 2020; Lan et al., 2019; de Vries et al., 2019).

Within the contrastive learning framework, the choice of loss function is significant. InfoNCE (Gutmann & Hyvärinen, 2010; Oord et al., 2018), which can be interpreted as a non-parametric estimation of the entropy of the representation (Wang & Isola, 2020), is a popular choice. Since then there have been a number of proposals (Zbontar et al., 2021; Grill et al., 2020; Hjelm et al., 2018); however, we show that InfoNCE is effective for our framework.

Recently, Yao et al. (2020) adapted the contrastive framework to large-scale recommendation systems in a way similar to our approach. The key difference is in the way the methods generate multiple views. Yao et al. (2020) proposes masking random features in a correlated manner and applying a dropout for categorical features, while our approach involves randomizing random features based on the features' respective empirical marginal distribution (in an uncorrelated way). Generating such views for a task is a difficult problem: there has been much work done in understanding and designing them (Wu et al., 2018; Purushwalkam & Gupta, 2020; Gontijo-Lopes et al., 2020; Lopes et al., 2019; Perez & Wang, 2017; Park et al., 2019) and learning them (Ratner et al., 2017; Cubuk et al., 2020; Ho et al., 2019; Lim et al., 2019; Zhang et al., 2019b; Tran et al., 2017; Tamkin et al., 2020).

Lastly, also similar to our work is VIME (Yoon et al., 2020), which proposes the same corruption technique for tabular data that we do. They pre-train an encoder network on unlabeled data by attaching "mask estimator" and "feature estimator" heads on top of the encoder state and teaching the model to recover both the binary mask that was used for corruption as well as the original uncorrupted input, given the corrupted input. The pre-trained encoder network is subsequently used for semi-supervised learning via attachment of a task-specific head and minimization of the supervised loss as well as an auto-encoder reconstruction loss. VIME was shown to achieve state-of-art results on genomics and clinical datasets. The key differences with our work is that we pre-train using a contrastive loss, which we show to be more effective than the denoising auto-encoder loss that partly constitutes VIME. Furthermore, after pre-training we fine-tune all model weights, including the encoder (unlike VIME, which only fine-tunes the task head), and we do so using task supervision only.

## 3 SCARF

We now describe our proposed method (Algorithm 1), which is also described in Figure 1. For each mini-batch of examples from the unlabeled training data, we generate a corrupted version $\tilde{x}^{(i)}$ for

each example $x^{(i)}$ as follows. We sample some fraction of the features uniformly at random and replace each of those features by a random draw from that feature's empirical marginal distribution, which is defined as the uniform distribution over the values that feature takes on across the training dataset. Then, we pass both $x^{(i)}$ and $\tilde{x}^{(i)}$ through the encoder network $f$, whose output we pass through the pre-train head network $g$, to get $z^{(i)}$ and $\tilde{z}^{(i)}$ respectively. Note that the pre-train head network $\ell_2$-normalizes the outputs so that they lie on the unit hypersphere – this has been found to be crucial in practice (Chen et al., 2020; Wang & Isola, 2020). We train on the InfoNCE contrastive loss, encouraging $z^{(i)}$ and $\tilde{z}^{(i)}$ to be close for all $i$ and $z^{(i)}$ and $\tilde{z}^{(j)}$ to be far apart for $i \neq j$, and we optimize over the parameters of $f$ and $g$ via SGD.

Then, to train a classifier for the task via fine-tuning, we take the encoder network $f$ and attach a classification head $h$ which takes the output of $f$ as its input and predicts the label of the example. We optimize the cross-entropy classification loss and tune the parameters of both $f$ and $h$.

While pre-training can be run for a pre-determined number of epochs, much like normal supervised training, how many is needed largely depends on the model and dataset. To this end, we propose using early stopping on the validation InfoNCE loss. Given unlabeled validation data, we cycle through it for some number of epochs, running our proposed method to generate $(x^{(i)}, \tilde{x}^{(i)})$ pairs. Once built, the loss on this static set is tracked during pre-training. Prototypical loss curves are shown in the Appendix.

## 4 EXPERIMENTS

We evaluate the impact of SCARF pre-training on test accuracy after supervised fine-tuning in three distinct settings: on the full dataset, on the full dataset but where only 25% of samples have labels and the remaining 75% do not, and on the full dataset where 30% of samples undergo label corruption.

**Datasets.** We use 69 datasets from the public OpenML-CC18 benchmark[1] under the CC-BY licence. It consists of 72 real-world classification datasets that have been manually curated for effective benchmarking. Since we're concerned with tabular datasets in this work, we remove MNIST, Fashion-MNIST, and CIFAR10. For each OpenML dataset, we form 70%/10%/20% train/validation/test splits, where a different split is generated for every trial and all methods use the same splits. The percentage used for validation and test are never changed and only training labels are corrupted for the label noise experiments.

**Dataset pre-processing**. We represent categorical features by a one-hot encoding, and most of the corruption methods explored in the ablations are on these one-hot encoded representations of the data (with the exception of SCARF, where the marginal sampling is done before one-hot encoding). We pre-process missing data as follows: if a feature column is always missing, we drop it. Otherwise, if the feature is categorical, we fill in missing entries with the mode, or most frequent, category computed over the full dataset. For numerical features, we impute it with the mean. We explore scaling numerical features by z-score, min-max, and mean normalization. We find that for a vanilla network (i.e. control), z-score normalization performed the best for all but three datasets (OpenML dataset ids 4134, 28, and 1468), for which no scaling was optimal. We thus do not scale these three datasets and z-score normalize all others.

**Model architecture and training**. Unless specified otherwise, we use the following settings across all experiments. As described earlier, we decompose the neural network into an encoder $f$, a pre-training head $g$, and a classification head $h$, where the inputs to $g$ and $h$ are the outputs of $f$. We choose all three component models to be ReLU networks with hidden dimension 256. $f$ consists of 4 layers, whereas both $g$ and $h$ have 2 layers. Both SCARF and the autoencoder baselines use $g$ (for both pre-training and co-training, described later), but for autoencoders, the output dimensionality is the input feature dimensionality, and the mean-squared error reconstruction loss is applied. We train all models and their components with the Adam optimizer using the default learning rate of 0.001. For both pre-training and fine-tuning we use 128 batch size. Unsupervised pre-training methods all use early stopping with patience 3 on the validation loss, unless otherwise noted. Supervised fine-tuning uses this same criterion (and validation split), but classification error is used as the validation metric for early stopping, as it performs slightly better. We set a max number of fine-tune epochs of 200 and

---

[1] https://docs.openml.org/benchmark/

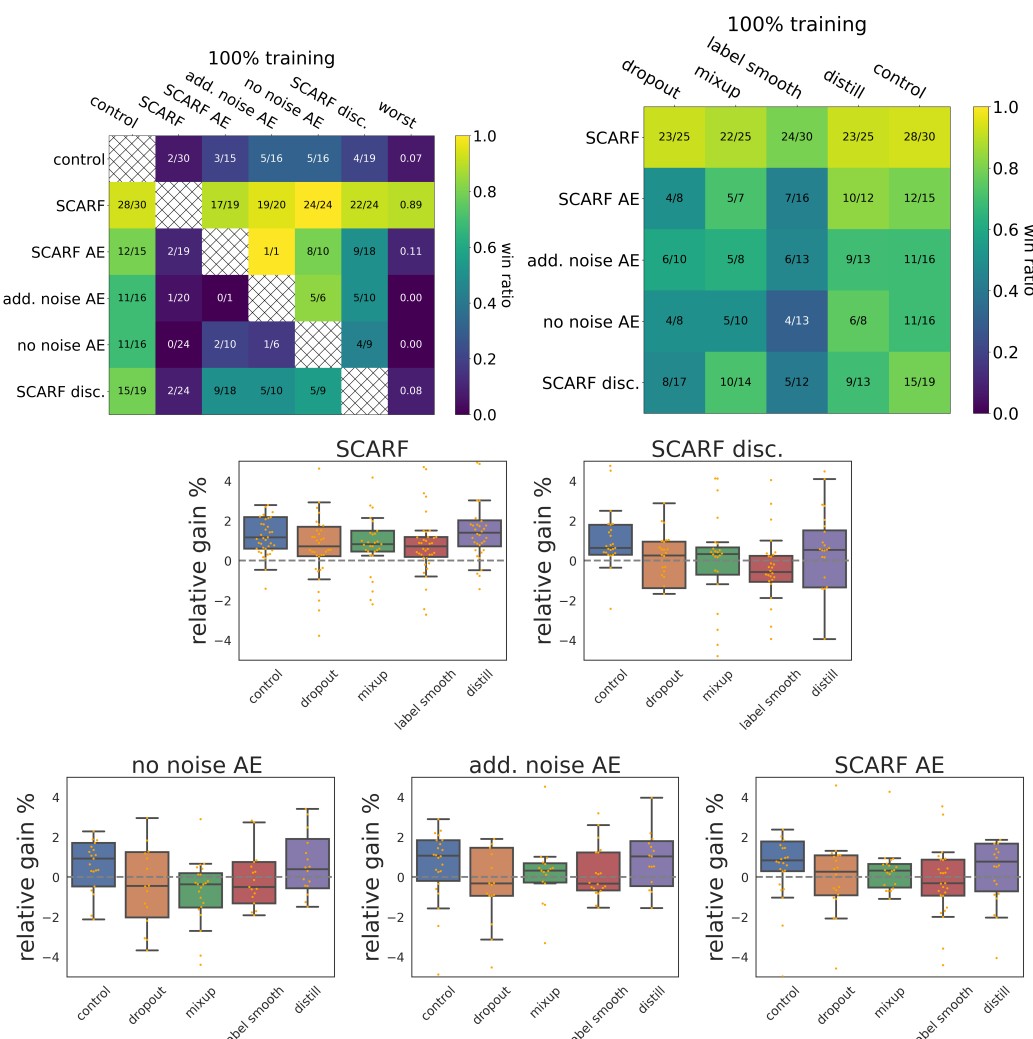

Figure 2: **Top**: Win matrices comparing pre-training methods against each other, and their improvement to existing solutions. **Bottom**: Box plots showing the relative improvement of different pre-training methods over baselines (y-axis is zoomed in). We see that SCARF pre-training adds value *even* when used in conjunction with known techniques.

pre-train epochs of 1000, We use 10 epochs to build the static validation set. Unless otherwise noted, we use a corruption rate $c$ of 0.6 and a temperature $\tau$ of 1, for SCARF-based methods. All runs are repeated 30 times using different train/validation/test splits. Experiments were run on a cloud cluster of CPUs, and we used about one million CPU core hours in total for the experiments.

**Evaluation methods**. We use the following to effectively convey the results across all datasets.

*Win matrix*. Given $M$ methods, we compute a "win" matrix $W$ of size $M \times M$, where the $(i, j)$ entry is defined as:

$$W_{i,j} = \frac{\sum_{d=1}^{69} \mathbb{1}[\text{method } i \text{ beats } j \text{ on dataset } d]}{\sum_{d=1}^{69} \mathbb{1}[\text{method } i \text{ beats } j \text{ on dataset } d] + \mathbb{1}[\text{method } i \text{ loses to } j \text{ on dataset } d]}.$$

"Beats" and "loses" are only defined when the means are not a statistical tie (using Welch's $t$-test with unequal variance and a $p$-value of 0.05). A win ratio of $0/1$ means that out of the 69 (pairwise) comparisons, only one was significant and it was a loss. Since $0/1$ and $0/69$ have the same value but the latter is more confident indication that $i$ is worse than $j$, we present the values in fractional form

and use a heat map. We add an additional column to the matrix that represents the minimum win ratio across each row.

*Box plots*. The win matrix effectively conveys how often one method beats another but does not capture the degree by which. To that end, for each method, we compute the relative percent improvement over some reference method on each dataset. We then build box-plots depicting the distribution of the relative improvement across datasets, plotting the observations as small points. We only consider datasets where the means of the method and the reference are different with $p$-value $0.20$. We use a larger $p$-value here than when computing the win ratio because otherwise some methods would not have enough points to make the box-plots meaningful.

**Baselines.** We use the following baselines:

- *Label smoothing* (Szegedy et al., 2016), which has proved successful for accuracy (Müller et al., 2019) and label noise (Lukasik et al., 2020). We use a weight of $0.1$ on the smoothing term.
- *Dropout*. We use standard dropout (Srivastava et al., 2014) using rate $0.04$ on all layers. Dropout has been shown to improve performance and robustness to label noise (Rusiecki, 2020).
- *Mixup* (Zhang et al., 2017), using $\alpha = 0.2$.
- *Autoencoders* (Rumelhart et al., 1985). We use this as our key ablative pre-training baseline. We use the classical autoencoder ("no noise AE"), the denoising autoencoder (Vincent et al., 2008; 2010) using Gaussian additive noise ("add. noise AE") as well as SCARF's corruption method ("SCARF AE"). We use MSE for the reconstruction loss. We try both pre-training and co-training with the supervised task, and when co-training, we add $0.1$ times the autoencoder loss to the supervised objective. We discuss co-training in the Appendix as it is less effective than pre-training.
- SCARF *data-augmentation*. In order to isolate the effect of our proposed feature corruption technique, we skip pre-training and instead train on the corrupted inputs during supervised fine-tuning. We discuss results for this baseline in the Appendix as it is less effective than the others.
- *Discriminative* SCARF. Here, our pre-training objective is to discriminate between original input features and their counterparts that have been corrupted using our proposed technique. To this end, we update our pre-training head network to include a final linear projection and swap the InfoNCE with a binary logistic loss. We use classification error, not logistic loss, as the validation metric for early stopping, as we found it to perform slightly better.
- *Self-distillation* (Hinton et al., 2015; Zhang et al., 2019a). We first train the model on the labeled data and then train the final model on both the labeled and unlabeled data using the first models' predictions as soft labels for both sets.
- *Deep $k$-NN* (Bahri et al., 2020), a recently proposed method for label noise. We set $k = 50$.
- *Bi-tempered loss* (Amid et al., 2019), a recently proposed method for label noise. We use $5$ iterations, $t_1 = 0.8$, and $t_2 = 1.2$.
- *Self-training* (Yarowsky, 1995; McClosky et al., 2006). A classical semi-supervised method – each iteration, we train on pseudo-labeled data (initialized to be the original labeled dataset) and add highly confident predictions to the training set using the prediction as the label. We then train our final model on the final dataset. We use a softmax prediction threshold of $0.75$ and run for $10$ iterations.
- *Tri-training* (Zhou & Li, 2005). Like self-training, but using three models with different initial labeled data via bootstrap sampling. Each iteration, every model's training set is updated by adding only unlabeled points whose predictions made by the other two models agree. It was shown to be competitive in modern semi-supervised NLP tasks (Ruder & Plank, 2018). We use same hyperparameters as self-training.

### 4.1 SCARF PRE-TRAINING IMPROVES PREDICTIVE PERFORMANCE

Figure 2 shows our results. From the first win matrix plot, we see that all five pre-training techniques considered improve over no pre-training (control), and that SCARF outperforms the others and has

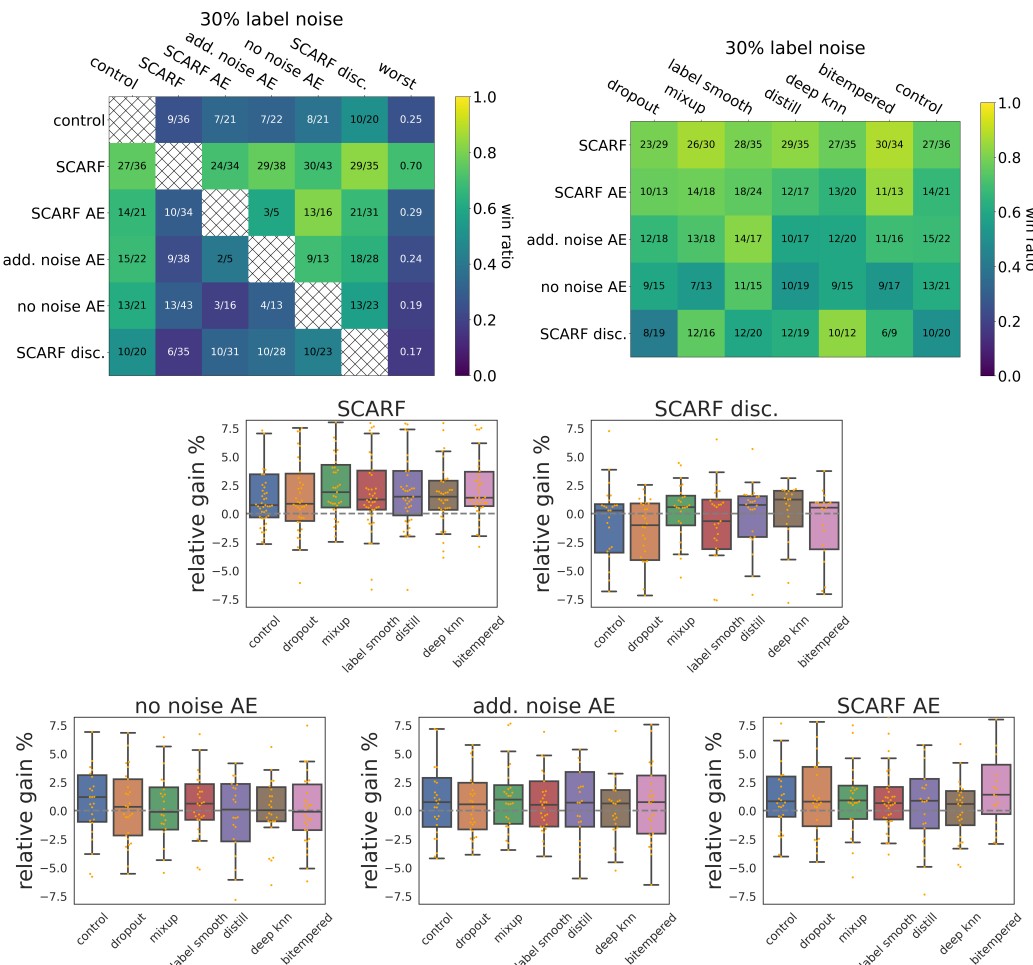

Figure 3: SCARF boosts baseline performance even when $30\%$ of the training labels are corrupted. Notably, it improves state-of-the-art label noise solutions like Deep $k$-NN.

more statistically significant wins. The second win matrix shows that SCARF pre-training boosts the performance of mixup, label smoothing, distillation, and dropout, and it does so better than alternatives. In other words, pre-training *complements* existing solutions, suggesting that a distinct mechanism is at play here. The box plots expand on the second win matrix, showing the relative improvement that each of the pre-training strategies confers over the baselines. Table 1 summarizes the box-plots by the average relative gain. We observe that SCARF generally outperforms the alternatives and adds a 1-2% relative gain across the board.

## 4.2 SCARF PRE-TRAINING IMPROVES PERFORMANCE IN THE PRESENCE TO LABEL NOISE

To show how pre-training improves model robustness when the data's labels are unreliable, we do as follows. Firstly, label noise robustness is often studied in two distinct settings - (1) when some subset of the training data is guaranteed to be uncorrupted and that set is known in advance, (2) when the entire dataset is untrustworthy. For simplicity, we consider setting 2 in our experiments. We corrupt labels as follows: leaving the validation and test splits uncorrupted, we select a random 30% percent of the training data to corrupt and for each datapoint, we replace its label by one of the classes, uniformly over all classes (including the datapoint's true class). Results are shown in Figure 3 and Table 1. Again, we see SCARF outperforms the rest and boosts all baselines by 2-3%.

| 100% labeled training | SCARF | SCARF AE | no noise AE | add. noise AE | SCARF disc. |
|---|---|---|---|---|---|
| control | **2.352** | 2.244 | 1.107 | 1.559 | 0.574 |
| dropout | **1.609** | 1.196 | 0.623 | 1.228 | -1.312 |
| mixup | **1.72** | 1.183 | -0.377 | 0.971 | -0.307 |
| label smooth | **1.522** | 0.711 | -0.002 | 1.04 | -0.894 |
| distill | **2.392** | 2.186 | 0.823 | 1.431 | -0.394 |
| *25% labeled training* | | | | | |
| control | **3.692** | 1.702 | 0.777 | 1.662 | 0.233 |
| dropout | **2.212** | 1.848 | 2.013 | 1.155 | -0.322 |
| mixup | **2.809** | 0.73 | 0.106 | 0.439 | 0.466 |
| label smooth | **2.303** | 0.705 | -0.564 | 0.196 | -0.206 |
| distill | **3.609** | 2.441 | 1.969 | 2.263 | 1.795 |
| self-train | **3.839** | 2.753 | 1.672 | 2.839 | 2.559 |
| tri-train | **3.549** | 2.706 | 1.455 | 2.526 | 1.92 |
| *30% label noise* | | | | | |
| control | **2.261** | 1.988 | 0.914 | 1.612 | -1.408 |
| dropout | 2.004 | **2.058** | 0.9 | 1.471 | -2.54 |
| mixup | **2.739** | 1.723 | 0.116 | 1.409 | 0.189 |
| label smooth | **2.558** | 1.474 | 0.703 | 1.395 | -1.337 |
| distill | **2.881** | 2.296 | -0.239 | 1.659 | -0.226 |
| deep knn | **2.001** | 1.281 | 0.814 | 1.348 | 0.088 |
| bitempered | 2.68 | **2.915** | 0.435 | 1.387 | -1.147 |

Table 1: Results using the fully labeled training data, only 25% of the labeled training data, and the full training data subject to 30% label noise. Shown is the average relative gain in accuracy when adding the pre-training methods (columns) to the reference methods (rows). Like the box-plots, we filter out datasets using $p$-value 0.20. We see that SCARF consistently outperforms alternatives, not only in improving control but also in improving methods designed specifically for the setting.

### 4.3 SCARF PRE-TRAINING IMPROVES PERFORMANCE WHEN LABELED DATA IS LIMITED

To show how pre-training helps when there is more unlabeled data than labeled ones, we remove labels in the training split so that only 25% of the original split remains labeled. Autoencoders, SCARF, self-training and tri-training all leverage the unlabeled remainder. Results are shown in Figure 4 (Appendix) and Table 1. SCARF outperforms the rest, adding a very impressive 2-4% to all baselines.

### 4.4 ABLATIONS

We now detail the importance of every factor in SCARF. We show only some of the results here; the rest are in the Appendix.

**Other corruption strategies are less effective and are more sensitive to feature scaling.** Here, we ablate the marginal sampling corruption technique we proposed, replacing it with the following other promising strategies, while keeping all else fixed.

1. *No corruption.* We do not apply any corruption - i.e. $\tilde{x}^{(i)} = x^{(i)}$, in Algorithm 1. In this case, the cosine similarity between positive pairs is always one and the model is learning to make negative pairs as orthogonal as possible. Under the recent perspective (Wang & Isola, 2020) that the contrastive loss comprises two terms – one that encourages alignment between views of the same example – and one that encourages the hypersphere embeddings to be uniformly spread out – we see that with no corruption, pre-training may just be learning to embed input examples uniformly on the hypersphere.

2. *Mean corruption.* After determining which features to corrupt, we replace their entries with the empirical marginal distribution's mean.

3. *Additive Gaussian noise.* We add i.i.d $\mathcal{N}(0, 0.5^2)$ noise to features.

4. *Joint sampling.* Rather than replacing features by random draws from their marginals to form $\tilde{x}^{(i)}$, we instead randomly draw $\hat{x}^{(i)}$ from training data $\mathcal{X}$ – i.e. we draw from the empirical (joint) data distribution – and then set $\tilde{x}_j^{(i)} = \hat{x}_j^{(i)} \ \forall j \in \mathcal{I}_i$.

5. *Missing feature corruption.* We mark the selected features as "missing" and add one learnable value per feature dimension to our model. When a feature is missing, it's filled in with the corresponding learnable missing value.

6. *Feature dropout.* We zero-out the selected features.

We also examine the corruption strategies under the following ways of scaling the input features.

1. *Z-score scaling.* Here, $x_j = [x_j - \text{mean}(\mathcal{X}_j)] / \text{std}(\mathcal{X}_j)$.

2. *Min-max scaling.* $x_j = [x_j - \min(\mathcal{X}_j)] / [\max(\mathcal{X}_j) - \min(\mathcal{X}_j)]$.

3. *Mean scaling.* $x_j = [x - \text{mean}(\mathcal{X}_j)] / [\max(\mathcal{X}_j) - \min(\mathcal{X}_j)]$.

Figure 6 (Appendix) shows the results for z-score and min-max scaling. SCARF marginal sampling generally outperforms the other corruption strategies for different types of feature scaling. Marginal sampling is neat in that in addition to not having hyperparameters, it is invariant to scaling and preserves the "units" of each feature. In contrast, even a simple multiplicative scaling requires the additive noise to be scaled in the same way.

**SCARF is *not* sensitive to batch size.** Contrastive methods like SimCLR (Chen et al., 2020) have shown consistent improvements upon increasing the batch size, $N$. There is a tight coupling between the batch size and how hard the contrastive learning task is, since, in our case, the loss term for each example $i$ involves 1 positive and $N-1$ negatives. The need for large (e.g. 5000) batch sizes has motivated engineering solutions to support them (Gao & Zhang, 2021) and have been seen as grounds for adopting other loss functions (Zbontar et al., 2021). Figure 5 (Appendix) compares a range of batch sizes. We see that increasing the batch size past 128 did not result in significant improvements. A reasonable hypothesis here is that higher capacity models and harder tasks benefit more from negatives.

**SCARF is fairly insensitive to the corruption rate and temperature.** We study the effect of the corruption rate $c$ in Figure 5 (Appendix). We see that performance is stable when the rate is in the range $50\% - 80\%$. We thus recommend a default setting of $60\%$. We see a similar stability with respect to the temperature hyperparameter (see Appendix). We recommend using a default temperature of 1.

**Tweaks to the corruption do not work any better.** The Appendix shows the effect of four distinct tweaks to the corruption method. We do not see any reason to use any of them.

**Alternatives to InfoNCE do not work any better.** We investigate the importance of our choice of InfoNCE loss function and see the effects of swapping it with recently proposed alternatives Alignment and Uniformity (Wang & Isola, 2020) and Barlow Twins (Zbontar et al., 2021). We found that these alternative losses almost match or perform worse than the original and popular InfoNCE in our setting. See the Appendix for details.

## 5 CONCLUSION

Self-supervised learning has seen profound success in important domains including computer vision and natural language processing, but little progress has been made for the general tabular setting. We propose a self-supervised learning method that's simple and versatile and learns representations that are effective in downstream classification tasks, even in the presence of limited labeled data or label noise. Potential negative side effects of this method may include learning representations that reinforce biases that appear in the input data. Finding ways to mitigate this during the training process is a potential direction for future research.

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

# A  APPENDIX

We now present findings held out from the main text. Unless noted otherwise, we use the same hyperparameters (i.e. $c = 0.6$, temperature $\tau = 1$, etc.).

## A.1  RESULTS FOR DATA-LIMITED EXPERIMENTS

Figure 4 presents the results for the case when only $25\%$ of the training labels are available. We see that SCARF outperforms the rest here as well.

## A.2  PRE-TRAINING LOSS CURVES

Figure 7 shows prototypical pre-training training and validation loss curves for SCARF. We see that both losses tend to decrease rapidly at the start of training and then diminish slowly until the early stopping point. We also notice the noisy (high variance) nature of the training loss - this is due to the stochasticity of our corruption method.

## A.3  ABLATIONS CONTINUED

We present more ablation results, where the metric is accuracy using $100\%$ of the labeled data.

### IMPACT OF BATCH SIZE AND CORRUPTION RATES

Figure 5 examines the impact of the batch size and corruption rate for the fully labeled, noiseless setting.

### IMPACT OF CORRUPTION STRATEGIES

Figure 6 shows the performance of a variety of alternative corruption strategies for SCARF under z-score and min-max feature normalization. Results for mean scaling are shown in Figure 12.

### IMPACT OF TEMPERATURE

Figure 8 shows the impact of the temperature term. While prior work considers temperature an important hyperparameter that needs to be tuned, we see that a default of 1 (i.e. just softmax) works the best in our setting.

### MORE CORRUPTION ABLATIONS

Figure 11 shows the following points.

- Corrupting one view is better than corrupting both the views. The likely explanation for this is that at a corruption rate of $60\%$, corrupting both views i.i.d means that the fraction of feature indices that contain the same value for both views is small - in other words, there is less information between the two views and the contrastive task is harder.

- Corrupting the same set of feature indices within the mini-batch performs slightly worse than random sampling for every example in the mini-batch.

- An alternative way to select feature indices to corrupt is to use Bernoulli's: for each feature index, corrupt it with probability (corruption rate) $c$, ensuring that at least one index is corrupted. In the selection method we describe in Algorithm 1, a constant number of indices are corrupted, whereas here it is variable. This alternative way of selecting indices performs roughly the same.

- Once the feature indices to corrupt are determined, rather than corrupting them by drawing from the empirical marginal distribution, we can instead draw a random example from the training set and use its feature entries for corrupting *all* views for the mini-batch. This performs worse.

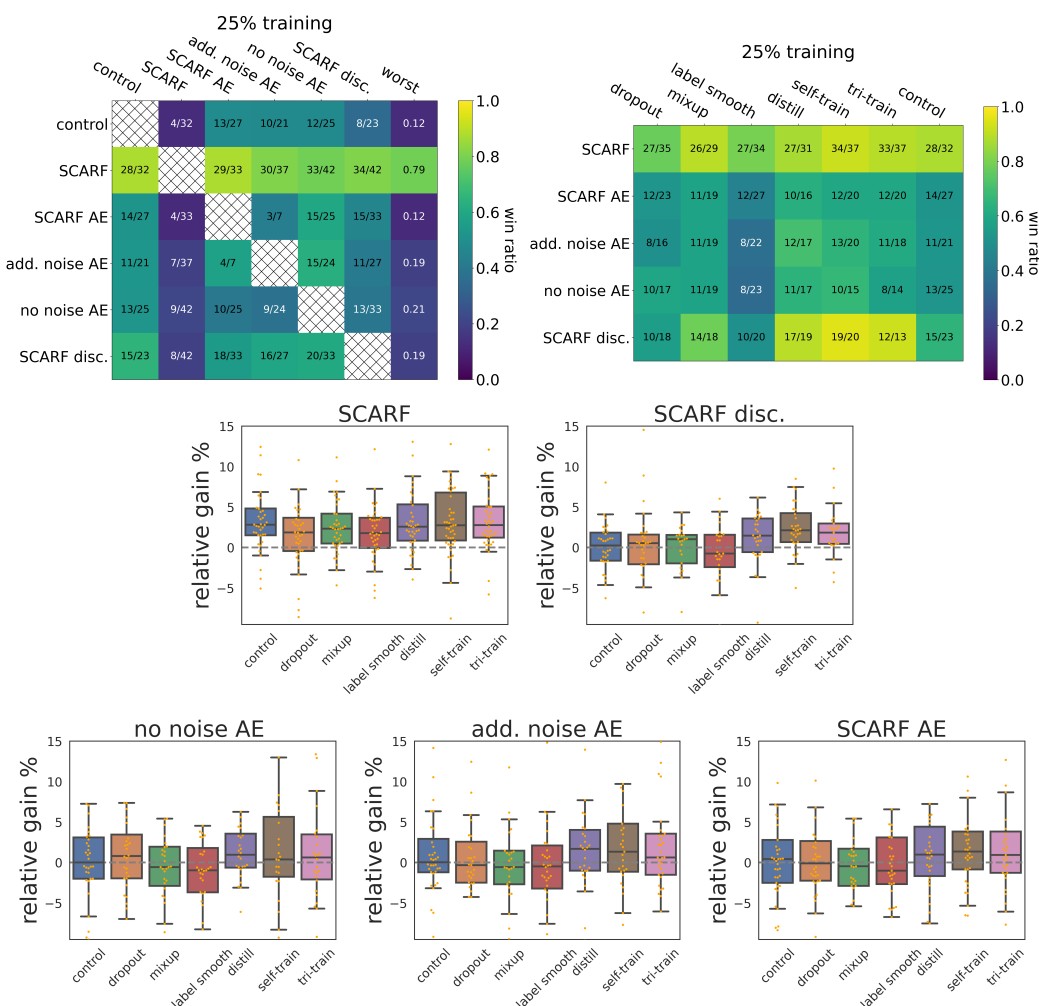

Figure 4: SCARF shows even more significant gain in the semi-supervised setting where 25% of the data is labeled and the remaining 75% is not. Strikingly, pre-training with SCARF boosts the performance of self-training and tri-training by several percent.

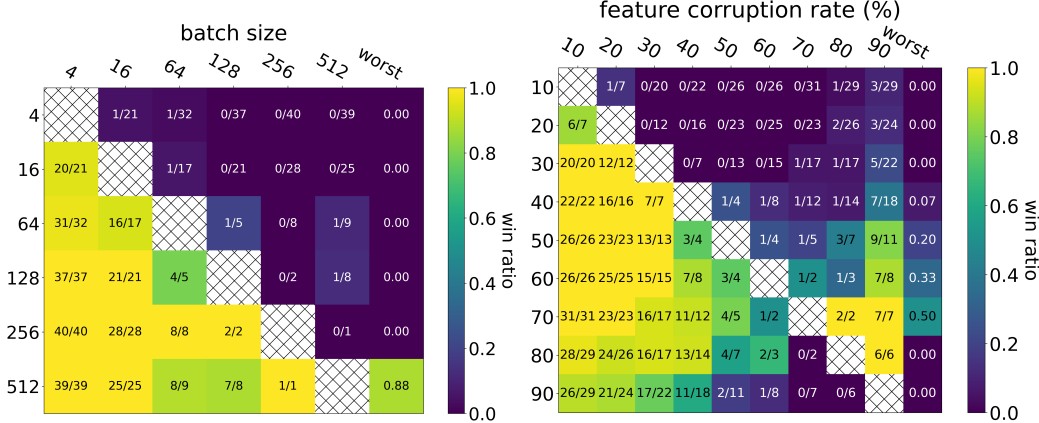

Figure 5: Win matrix for various batch sizes (**Left**) and corruption rates (**Right**) for the fully labeled, noiseless setting.

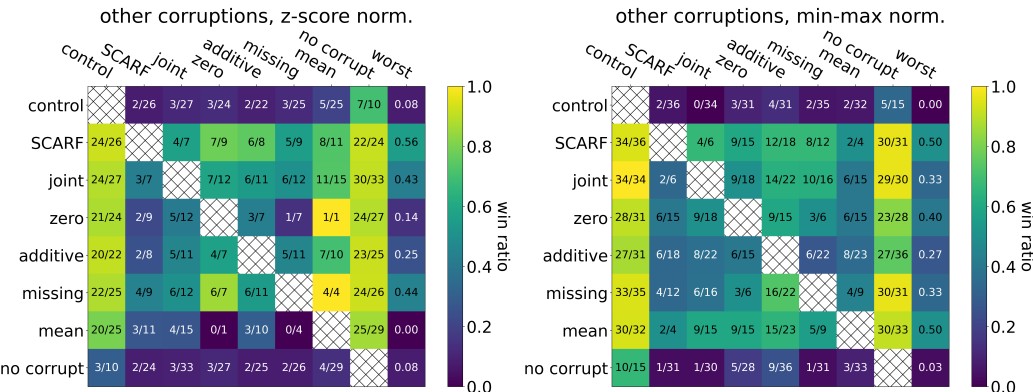

Figure 6: **Left:** Win matrix comparing different corruption strategies when z-score feature normalization is used in the fully labeled, noiseless setting. **Right:** The same matrix but when min-max feature scaling is used. We see that SCARF is better than alternative corruption strategies for different types of feature scaling.

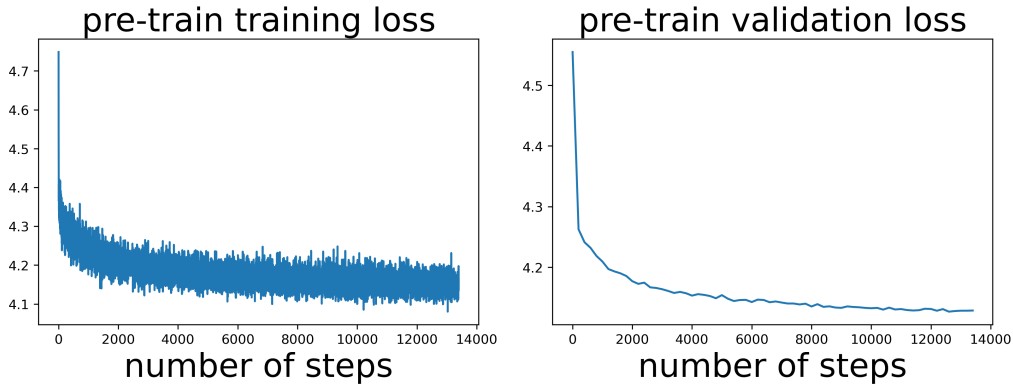

Figure 7: Training and validation loss curves for SCARF pre-training on Phonemes (dataset id 1489). We observe that both losses drop rapidly initially and then taper off. The training curve is jittery because of the random corruptions, but the validation curve isn't because the validation dataset is built once before training and is static throughout training.

ALTERNATIVE LOSSES

Figure 8 shows the impact of using two recently-proposed alternatives to the InfoNCE loss: Uniform and Align (Wang & Isola, 2020) and Barlow Twins (Zbontar et al., 2021). We use 5e-3 for the hyperparameter in Barlow, and equal weighting between the align and uniform loss terms. We find no benefit in using these other losses.

SCARF PRE-TRAINING OUTPERFORMS SCARF CO-TRAINING

Here we address the question: is it better to pre-train using SCARF or to apply it as a term in the supervised training loss? In particular, $\mathcal{L}_{\text{co-train}} = \mathcal{L}_{\text{supervised}} + \lambda_{\text{cont}}\mathcal{L}_{\text{cont}}$, where $\mathcal{L}_{\text{cont}}$ is as described in Algorithm 1. Figure 9 shows that pre-training outperforms co-training for a range of different $\lambda_{\text{cont}}$. We see this is also the case for additive noise autoencoders.

SCARF PRE-TRAINING OUTPERFORMS SCARF DATA AUGMENTATION

Figure 10 shows that using SCARF only for data augmentation during supervised training performs far worse than using it for pre-training.

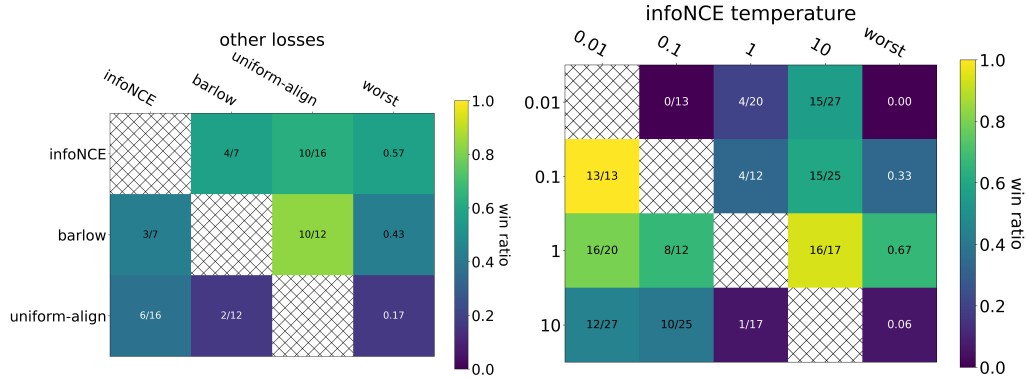

Figure 8: **Left**: Barlow Twins loss performs similar to InfoNCE while Uniform-Align performs worse. **Right**: InfoNCE softmax temperature 1 performs well.

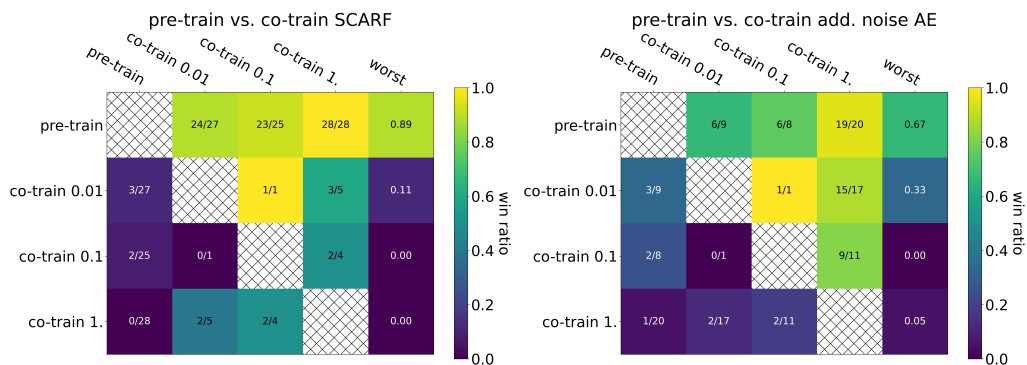

Figure 9: **Left:** Pre-training with SCARF beats co-training for a range of weights on the co-training term. **Right:** The same is true for additive noise autoencoders.

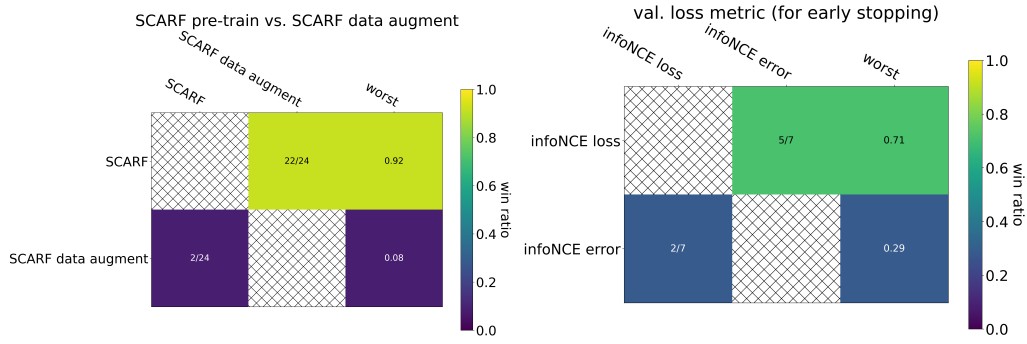

Figure 10: **Left**: Using SCARF only for data augmentation during supervised training performs worse than using it for pre-training. **Right**: Using InfoNCE error instead of InfoNCE loss as the validation metric for early stopping degrades performance.

USING INFONCE ERROR AS THE PRE-TRAINING VALIDATION METRIC IS WORSE

SCARF uses the same loss function (InfoNCE) for training and validation. If we instead use the InfoNCE error for validation - where an error occurs when the model predicts an off-diagonal entry of the batch-size by batch-size similarity matrix - downstream performance degrades, as shown in Figure 10.

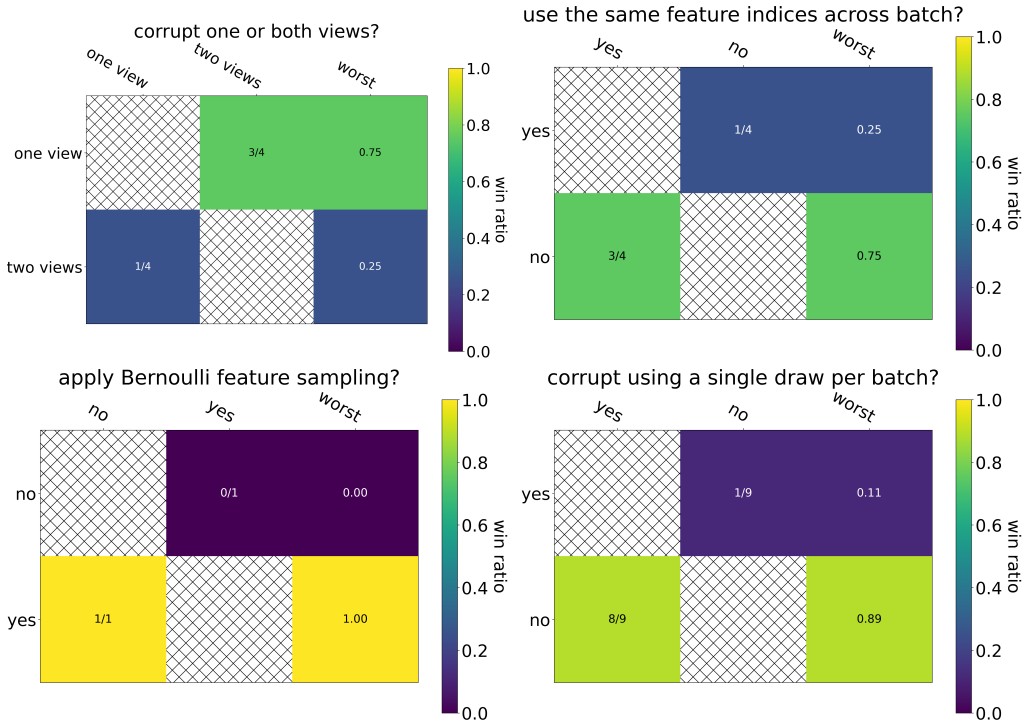

Figure 11: **Top left**: Corrupting one view is better than corrupting both. **Top right**: Using different random feature indices for each example in the mini-batch is better than using a same set across the batch. **Bottom left**: Selecting a variable number of feature indices via coin flips performs similar to the method described in Algorithm 1. **Bottom right**: Corrupting by replacing the features by the features of a *single* drawn example for the whole mini-batch performs worse.

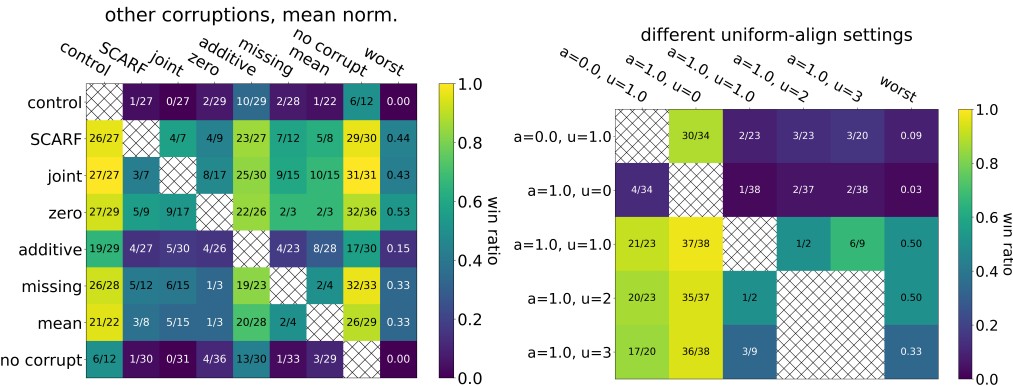

Figure 12: **Left:** SCARF's corruption strategy remains competitive for "mean" feature scaling, as was the case for both z-score and min-max scaling. **Right:** Comparison of different hyperparameters for the uniform-align loss. $a$ and $u$ are the weights on the align and uniform terms respectively.

MORE ON UNIFORM-ALIGN LOSS

Figure 12 compares SCARF using the uniform-align loss, for different weights between the align and uniform loss terms. The best performance is achieved with equal weight between the two, and Figure 8 shows that this underperforms vanilla InfoNCE. We thus recommend using the vanilla InfoNCE loss.

ABSOLUTE PERFORMANCE

We have thus far presented results in terms of relative improvement over control since relative gain is easier to summarize across the many datasets we consider. In this section, we present the absolute test accuracy for all methods and datasets, for the keen reader. We also add a gradient-boosted decision tree baseline using the XGBoost package, since decision trees are common among practitioners working with tabular data. We use the Python API [2], version 0.6, and choose the default settings for XGBClassifier (max depth of 3, 100 estimators, learning rate of 0.1).

Tables 2, 3, 4 show the absolute test accuracies (averaged over the 30 trials) for each dataset and baseline for the 100% training data, the 30% label noise, and the 25% (semi-supervised) training data settings respectively.

TRAINING SPEED

We use an early stopping criteria when fine-tuning SCARF and baselines. Table 5 shows the number of actual training epochs used (averaged over the 30 trials) for each dataset and baseline for the 100% training data setting.

---

[2]https://xgboost.readthedocs.io/en/latest/python/python_api.html

| dataset id | control | SCARF | SCARF AE | add. noise AE | no noise AE | SCARF disc. | XGBoost |
|---|---|---|---|---|---|---|---|
| 6 | 94.69 | 95.75 | 94.6 | 94.56 | 94.36 | 95.1 | 88.01 |
| 6332 | 74.24 | 75.4 | 71.76 | 69.35 | 66.6 | 69.34 | 77.59 |
| 54 | 75.45 | 76.75 | 75.14 | 75.83 | 75.88 | 78.44 | 76.16 |
| 50 | 90.54 | 97.53 | 98.68 | 97.33 | 95.91 | 96.66 | 90.61 |
| 46 | 93.47 | 94.59 | 94.63 | 94.96 | 94.33 | 93.36 | 95.72 |
| 469 | 18.54 | 19.08 | 19.89 | 19.14 | 19.79 | 19.17 | 21.21 |
| 458 | 98.96 | 99.51 | 99.34 | 99.2 | 99.11 | 99.19 | 98.46 |
| 4538 | 56.28 | 60.32 | 56.78 | 56.89 | 56.84 | 57.06 | 56.31 |
| 4534 | 96.08 | 95.85 | 95.91 | 95.8 | 95.9 | 96.46 | 94.66 |
| 44 | 93.4 | 94.46 | 93.53 | 93.45 | 93.49 | 93.55 | 94.54 |
| 4134 | 75.53 | 75.51 | 76.25 | 75.52 | 75.77 | 75.47 | 78.31 |
| 41027 | 90.96 | 91.13 | 91.73 | 92.15 | 92.32 | 91.68 | 81.75 |
| 40994 | 90.97 | 90.91 | 91.02 | 91.5 | 90.99 | 90.86 | 94.29 |
| 40984 | 90.56 | 91.97 | 90.32 | 90.46 | 90.53 | 90.69 | 92.66 |
| 40983 | 98.59 | 98.7 | 98.64 | 98.66 | 98.66 | 98.77 | 98.6 |
| 40982 | 73.48 | 73.27 | 73.47 | 73.7 | 73.56 | 72.99 | 78.07 |
| 40979 | 96.68 | 97.45 | 96.36 | 96.39 | 96.6 | 96.59 | 95.91 |
| 40978 | 94.84 | 95.13 | 95.36 | 95.81 | 94.75 | 94.74 | 97.33 |
| 40975 | 95.27 | 97.26 | 96.86 | 97.02 | 96.76 | 96.7 | 95.87 |
| 40966 | 98.44 | 98.77 | 98.21 | 98.46 | 98.68 | 97.87 | 97.05 |
| 40923 | 2.17 | 2.19 | 2.19 | 2.19 | 2.19 | 2.19 | 77.49 |
| 40701 | 91.73 | 92.36 | 92.36 | 92.27 | 92.14 | 92.29 | 95.31 |
| 40670 | 92.37 | 92.88 | 92.88 | 93.27 | 92.25 | 92.53 | 96.27 |
| 40668 | 83.38 | 82.97 | 83.29 | 83.17 | 83.26 | 83.17 | 74.11 |
| 40499 | 99.05 | 98.9 | 98.65 | 98.64 | 98.95 | 98.63 | 97.58 |
| 3 | 97.55 | 98.53 | 97.72 | 97.65 | 97.94 | 98.06 | 97.6 |
| 38 | 97.14 | 97.75 | 97.09 | 97.19 | 97.18 | 97.45 | 98.42 |
| 37 | 75.21 | 75.36 | 75.15 | 75.58 | 75.57 | 74.58 | 76.47 |
| 32 | 98.95 | 99.25 | 98.81 | 98.82 | 98.89 | 98.9 | 98.39 |
| 31 | 73.55 | 73.86 | 73.3 | 72.68 | 72.92 | 71.92 | 76.38 |
| 307 | 94.89 | 96.45 | 94.29 | 95.17 | 94.17 | 94.44 | 86.48 |
| 300 | 94.46 | 94.37 | 94.3 | 94.44 | 94.18 | 94.47 | 94.73 |
| 29 | 85.82 | 86.35 | 85.62 | 85.57 | 85.14 | 83.7 | 86.16 |
| 28 | 97.56 | 97.88 | 97.63 | 97.76 | 97.44 | 97.55 | 97.29 |
| 23 | 54.64 | 54.25 | 53.22 | 53.27 | 53.03 | 53.41 | 55.77 |
| 23517 | 51.59 | 51.55 | 51.53 | 51.51 | 51.69 | 51.57 | 52 |
| 23381 | 60.59 | 59.07 | 56.78 | 56.68 | 53.22 | 54.62 | 59.7 |
| 22 | 82.05 | 80.86 | 81.18 | 81.29 | 81.48 | 82.05 | 77.43 |
| 18 | 72.59 | 74.06 | 73.51 | 72.94 | 73.69 | 73.06 | 72.12 |
| 188 | 64.24 | 65.02 | 67.34 | 66.09 | 64.95 | 64.7 | 64.89 |
| 182 | 89.21 | 90.43 | 89.15 | 89.17 | 88.98 | 89.42 | 89.63 |
| 16 | 95.68 | 96.55 | 95.51 | 95.53 | 95.47 | 95.14 | 94.85 |
| 15 | 96.45 | 97.15 | 97.06 | 96.39 | 96.04 | 96.07 | 95.83 |
| 1590 | 85.55 | 85.6 | 85.51 | 85.51 | 85.54 | 85.5 | 86.45 |
| 151 | 82.77 | 83.55 | 82.69 | 82.67 | 82.34 | 83.18 | 83.27 |
| 1510 | 96.94 | 96.49 | 96.27 | 96.54 | 96.21 | 96.58 | 96.26 |
| 1501 | 91.41 | 93.76 | 91.55 | 91.6 | 92.49 | 92.35 | 91.33 |
| 14 | 79.27 | 80.99 | 81.05 | 80.67 | 80.89 | 78.82 | 82.34 |
| 1497 | 88.73 | 93.65 | 89.19 | 88.51 | 88.94 | 89.21 | 99.69 |
| 1494 | 86.76 | 86.71 | 86.96 | 86.59 | 87.29 | 86.44 | 86.32 |
| 1489 | 85.25 | 87.55 | 85.84 | 85.86 | 85.03 | 87.15 | 85.63 |
| 1487 | 93.47 | 94.25 | 94.27 | 94.31 | 94.27 | 93.96 | 94.34 |
| 1486 | 95.74 | 95.66 | 95.74 | 95.79 | 95.76 | 95.74 | 95.63 |
| 1485 | 54.2 | 70.16 | 68.76 | 68.24 | 64.12 | 56.73 | 71.6 |
| 1480 | 70.16 | 69.74 | 69.49 | 69.19 | 69.23 | 69.64 | 68.01 |
| 1478 | 96.85 | 97.16 | 97.08 | 97.12 | 97.07 | 94.98 | 98.38 |
| 1475 | 54.23 | 54.54 | 54.02 | 53.85 | 53.7 | 53.97 | 55.72 |
| 1468 | 93.45 | 94.04 | 92.95 | 91.68 | 92.72 | 85.79 | 91.27 |
| 1464 | 79.29 | 78.53 | 79.31 | 78.6 | 79.13 | 78.04 | 77.58 |
| 1462 | 99.5 | 99.94 | 99.75 | 99.72 | 99.77 | 99.92 | 99.49 |
| 1461 | 90.42 | 90.46 | 90.43 | 90.43 | 90.41 | 90.45 | 90.36 |
| 12 | 96.82 | 96.46 | 95.96 | 95.99 | 95.97 | 96.42 | 96.17 |
| 11 | 95.98 | 97.27 | 96.87 | 97.15 | 97.05 | 97.16 | 87.95 |
| 1068 | 93.04 | 93.18 | 93.23 | 93.06 | 93.12 | 93.18 | 93.78 |
| 1067 | 85.6 | 85.69 | 85.66 | 85.57 | 85.45 | 85.46 | 85.36 |
| 1063 | 84.17 | 84.06 | 84.06 | 83.71 | 82.16 | 83.7 | 83.27 |
| 1053 | 81.47 | 81.19 | 81.28 | 81.21 | 81.27 | 81.19 | 81.44 |
| 1050 | 89.84 | 89.49 | 89.24 | 89.27 | 89.4 | 89.7 | 88.9 |
| 1049 | 90.65 | 89.97 | 90.2 | 90.38 | 90.74 | 90.17 | 91.2 |

Table 2: Absolute test accuracy (percent; averaged over 30 trials) for the 100% training data setting, for each dataset and every baseline, including a gradient-boosted decision tree (XGBoost) baseline (max depth of 3, 100 estimators).

| dataset id | control | SCARF | SCARF AE | add. noise AE | no noise AE | SCARF disc. | XGBoost |
|---|---|---|---|---|---|---|---|
| 6 | 90.45 | 90.57 | 90.02 | 90.01 | 89.65 | 90.7 | 85.53 |
| 6332 | 69.39 | 70.99 | 63.41 | 62.78 | 60.25 | 62.96 | 72.75 |
| 54 | 72.21 | 71.94 | 70.84 | 71.11 | 71.93 | 72.76 | 72.92 |
| 50 | 79.64 | 87.66 | 90.52 | 86.7 | 85.14 | 88.12 | 87.4 |
| 46 | 84.43 | 92.1 | 92.76 | 93.05 | 91.89 | 85.77 | 94 |
| 469 | 19.47 | 19.02 | 20.08 | 19.89 | 19.41 | 18.14 | 19.52 |
| 458 | 96.65 | 99.27 | 98.06 | 97.97 | 97.1 | 95.47 | 96.07 |
| 4538 | 52.24 | 54.25 | 52.07 | 52.15 | 51.94 | 52.35 | 54.88 |
| 4534 | 94.13 | 94.38 | 94.2 | 94.09 | 94 | 94.31 | 93.66 |
| 44 | 91.37 | 93.21 | 91.67 | 91.34 | 91.31 | 91.76 | 93.13 |
| 4134 | 72.09 | 71.17 | 71.99 | 72.13 | 72.2 | 72.31 | 76.95 |
| 41027 | 85.54 | 85.31 | 85.43 | 85.42 | 85.46 | 85.53 | 81.54 |
| 40994 | 91.33 | 91.37 | 90.74 | 91.16 | 91.53 | 91.91 | 90.12 |
| 40984 | 88.63 | 89.99 | 88.44 | 88.51 | 88.08 | 88.68 | 90.71 |
| 40983 | 97.87 | 98.5 | 98.3 | 98.14 | 98.07 | 98.57 | 97.86 |
| 40982 | 70.54 | 70.92 | 71.26 | 70.97 | 71.11 | 70.19 | 73.69 |
| 40979 | 92.68 | 96.31 | 92.62 | 92.7 | 92.57 | 93.76 | 92.43 |
| 40978 | 91.87 | 92.77 | 93.81 | 94.12 | 93.64 | 92.52 | 96.48 |
| 40975 | 88.64 | 90.26 | 91.42 | 90.91 | 91.27 | 88.54 | 91.36 |
| 40966 | 90.78 | 88.52 | 88.65 | 90.12 | 91.86 | 87.37 | 89.58 |
| 40923 | 2.16 | 2.17 | 2.17 | 2.17 | 2.17 | 2.17 | 75.47 |
| 40701 | 88.77 | 89.34 | 88.87 | 88.54 | 88.92 | 89.41 | 93.7 |
| 40670 | 84.41 | 90.34 | 89.6 | 90.39 | 87.63 | 87.66 | 93.94 |
| 40668 | 80.62 | 80.27 | 80.66 | 80.7 | 80.71 | 80.55 | 73.67 |
| 40499 | 98.18 | 97.85 | 97.82 | 98.11 | 98.16 | 98.29 | 95.69 |
| 3 | 94.07 | 94.77 | 94.93 | 95.14 | 95.21 | 93.59 | 96.07 |
| 38 | 96.2 | 96.88 | 96.1 | 96.19 | 95.9 | 96.3 | 97.81 |
| 37 | 75.68 | 74.6 | 73.99 | 73.8 | 74.27 | 73.17 | 72.71 |
| 32 | 98.47 | 98.6 | 98.43 | 98.51 | 98.43 | 98.56 | 97.12 |
| 31 | 71.15 | 72.12 | 70.66 | 71.04 | 70.63 | 66.98 | 73 |
| 307 | 78.65 | 81.37 | 80.8 | 79.83 | 78.7 | 78.07 | 74.95 |
| 300 | 89.13 | 87.21 | 89.1 | 89.21 | 88.9 | 89.2 | 90.89 |
| 29 | 84.12 | 83.21 | 84.28 | 83.86 | 84.18 | 81.53 | 82.44 |
| 28 | 95.77 | 96.54 | 96.1 | 95.94 | 95.77 | 95.45 | 95.22 |
| 23 | 51 | 51.89 | 50.29 | 50.03 | 50.21 | 51.37 | 53.5 |
| 23517 | 51.33 | 51.12 | 51.29 | 51.25 | 51.38 | 51.26 | 51.7 |
| 23381 | 58.07 | 57.88 | 55.73 | 55.73 | 54.7 | 53.25 | 55.47 |
| 22 | 78.75 | 78.05 | 78.56 | 78.86 | 79.15 | 78.8 | 74.94 |
| 18 | 70.8 | 73.25 | 72.63 | 71.93 | 73.1 | 72.72 | 71.23 |
| 188 | 58.3 | 53.19 | 58.02 | 57.64 | 57.09 | 55.3 | 60.29 |
| 182 | 88.31 | 88.88 | 87.82 | 87.83 | 87.36 | 87.59 | 88.25 |
| 16 | 88.19 | 94.62 | 92.02 | 91.56 | 89.75 | 87.96 | 89.57 |
| 15 | 96.31 | 97.2 | 96.77 | 96.17 | 95.56 | 95.87 | 93.6 |
| 1590 | 84.81 | 85.19 | 85.07 | 85.06 | 85.06 | 85.05 | 86.23 |
| 151 | 80.08 | 81.15 | 80.06 | 80.14 | 80.02 | 80.52 | 82.97 |
| 1510 | 94.58 | 94.37 | 92.51 | 92.24 | 92.18 | 93.83 | 92.13 |
| 1501 | 78.73 | 89.29 | 81.16 | 81.73 | 82.16 | 84.43 | 85.6 |
| 14 | 72.34 | 78.57 | 77.87 | 77.52 | 74.92 | 71.53 | 77.64 |
| 1497 | 82.85 | 90.2 | 82.64 | 82.24 | 82.82 | 82.72 | 98.75 |
| 1494 | 84.18 | 84.97 | 85.03 | 85.06 | 85.18 | 84.72 | 83.4 |
| 1489 | 83.29 | 85.39 | 84.16 | 84.11 | 83.48 | 84.62 | 84.26 |
| 1487 | 92.61 | 92.97 | 93.27 | 93.41 | 92.9 | 93.16 | 93.55 |
| 1486 | 94.89 | 94.71 | 94.91 | 94.95 | 94.91 | 94.96 | 95.12 |
| 1485 | 53.97 | 67.03 | 66.74 | 66.62 | 62.55 | 54.21 | 65.67 |
| 1480 | 71.63 | 69.72 | 68.86 | 68.95 | 68.9 | 69.53 | 68.06 |
| 1478 | 94.13 | 94.42 | 94.75 | 94.71 | 94.71 | 94.74 | 96.37 |
| 1475 | 51.47 | 51.9 | 51.19 | 51.29 | 51.12 | 51.28 | 53.8 |
| 1468 | 83.07 | 90.76 | 83.39 | 79.58 | 78.46 | 64.89 | 87.3 |
| 1464 | 76.65 | 76.91 | 77.69 | 76.9 | 78.28 | 76.63 | 75.71 |
| 1462 | 98.97 | 99.55 | 99.14 | 99.21 | 99.52 | 99.69 | 98.07 |
| 1461 | 90.06 | 90.21 | 90.11 | 90.11 | 90.11 | 90.05 | 90.37 |
| 12 | 93.99 | 95.18 | 93.93 | 93.77 | 93.8 | 94.2 | 92.98 |
| 11 | 90.25 | 88.77 | 89.77 | 89.11 | 89.88 | 89.68 | 82.43 |
| 1068 | 92.45 | 92.44 | 92.32 | 92.24 | 92.1 | 92.73 | 92.82 |
| 1067 | 84.74 | 84.87 | 84.96 | 84.53 | 84.62 | 84.33 | 85.33 |
| 1063 | 83.81 | 83.76 | 83.59 | 83.63 | 82.95 | 83.95 | 80.95 |
| 1053 | 81.05 | 81.05 | 81.06 | 81.04 | 81.09 | 81.04 | 81.07 |
| 1050 | 89.08 | 88.94 | 88.96 | 89.11 | 89.46 | 89.32 | 88.81 |
| 1049 | 88.95 | 87.48 | 88.53 | 88.69 | 88.93 | 88.58 | 89.2 |

Table 3: Absolute test accuracy (percent; averaged over 30 trials) for the 30% label noise setting, for each dataset and every baseline, including a gradient-boosted decision tree (XGBoost) baseline (max depth of 3, 100 estimators).

| dataset id | control | SCARF | SCARF AE | add. noise AE | no noise AE | SCARF disc. | XGBoost |
|---|---|---|---|---|---|---|---|
| 6 | 87.22 | 89.71 | 87.12 | 87.38 | 87.01 | 88.87 | 85.02 |
| 6332 | 63.67 | 66.71 | 58.53 | 57.82 | 57.25 | 61.5 | 69.41 |
| 54 | 69.27 | 66.59 | 68.31 | 68.48 | 69.5 | 68.85 | 69.04 |
| 50 | 72.63 | 79.2 | 86.85 | 82.9 | 75.78 | 88.62 | 81.34 |
| 46 | 87.14 | 93.1 | 93.2 | 93.86 | 93.28 | 89.35 | 94.6 |
| 469 | 18.02 | 18.53 | 19.31 | 19.16 | 19.11 | 17.85 | 18.81 |
| 458 | 98.03 | 99.48 | 98.6 | 98.21 | 97.54 | 98.16 | 95.58 |
| 4538 | 50.65 | 53.75 | 50.79 | 50.32 | 50.45 | 50.03 | 54.05 |
| 4534 | 94.1 | 94.31 | 93.89 | 94.06 | 93.93 | 94.3 | 94.22 |
| 44 | 91.6 | 93.6 | 92 | 91.95 | 91.63 | 92.5 | 93.41 |
| 4134 | 68.31 | 68.44 | 69.65 | 69 | 68.78 | 68.45 | 74.58 |
| 41027 | 84.3 | 84.63 | 84.7 | 84.66 | 84.68 | 85.24 | 80.84 |
| 40994 | 90.63 | 90.96 | 90.96 | 90.93 | 90.97 | 90.96 | 92.69 |
| 40984 | 86.48 | 88.05 | 86.12 | 85.83 | 86.34 | 86.49 | 89.96 |
| 40983 | 94.72 | 97.65 | 94.85 | 94.65 | 97.72 | 98.58 | 97.92 |
| 40982 | 68.25 | 67.99 | 67.62 | 68.08 | 68.62 | 67.11 | 70.14 |
| 40979 | 92.81 | 96.21 | 92.58 | 92.15 | 92.56 | 93.55 | 91.33 |
| 40978 | 86.05 | 89.95 | 91.12 | 91.49 | 88.9 | 85.88 | 96.32 |
| 40975 | 72.47 | 90.55 | 91.14 | 90.53 | 90.02 | 71.68 | 91.98 |
| 40966 | 86.53 | 84.11 | 83.98 | 86.01 | 88.3 | 83.48 | 80.65 |
| 40923 | 2.16 | 2.16 | 2.16 | 2.16 | 2.16 | 2.16 | 74.35 |
| 40701 | 87.44 | 89.59 | 88.76 | 88.49 | 88.44 | 88.34 | 93.46 |
| 40670 | 81.67 | 90.97 | 89.69 | 90.27 | 87.58 | 88.22 | 94.37 |
| 40668 | 79.64 | 79.28 | 79.41 | 79.72 | 79.74 | 79.37 | 74.05 |
| 40499 | 97.88 | 97.65 | 97.13 | 97.59 | 98.13 | 97.78 | 94.59 |
| 3 | 93.04 | 95.44 | 94.02 | 94.84 | 94.34 | 94.59 | 96.16 |
| 38 | 95.13 | 96.92 | 95.39 | 95.17 | 94.53 | 95.39 | 97.7 |
| 37 | 70.37 | 72.24 | 71.15 | 71.2 | 70.21 | 71.58 | 72.94 |
| 32 | 98.39 | 98.61 | 98.19 | 98.15 | 97.94 | 98.4 | 96.9 |
| 31 | 70.53 | 71.66 | 68.48 | 68.54 | 65.82 | 69.33 | 73.12 |
| 307 | 69.68 | 72.47 | 73.49 | 72.6 | 72.31 | 70.88 | 63.97 |
| 300 | 91.66 | 87.01 | 89.98 | 90.42 | 90.58 | 91.06 | 90.3 |
| 29 | 83.6 | 83.57 | 80.7 | 81.46 | 81.93 | 82.28 | 84.59 |
| 28 | 96.43 | 96.56 | 96.38 | 96.08 | 96.23 | 96.12 | 95.07 |
| 23 | 49.56 | 51.45 | 45.64 | 46.48 | 44.84 | 49.95 | 51.48 |
| 23517 | 51.17 | 51.04 | 51.03 | 50.98 | 51.03 | 51.23 | 51.52 |
| 23381 | 58.83 | 58.7 | 55.62 | 55.37 | 53.37 | 56.43 | 55.5 |
| 22 | 77.72 | 77.03 | 76.27 | 76.8 | 77.8 | 76.34 | 73.72 |
| 18 | 70.71 | 71.37 | 71.05 | 70.53 | 72.39 | 71.11 | 70.11 |
| 188 | 55.18 | 46.41 | 50.56 | 53.42 | 53.06 | 51.72 | 57.82 |
| 182 | 86.98 | 88.36 | 87.37 | 87.34 | 86.53 | 86.62 | 87.89 |
| 16 | 88.25 | 93.91 | 90.7 | 90.44 | 89.6 | 89.97 | 87.54 |
| 15 | 96.65 | 97.01 | 96.7 | 96.45 | 94.54 | 95.54 | 95.19 |
| 1590 | 84.9 | 85.01 | 84.89 | 84.89 | 84.92 | 84.86 | 86.34 |
| 151 | 79.49 | 80.7 | 79.48 | 79.61 | 79.4 | 80.13 | 82.68 |
| 1510 | 94.63 | 94.36 | 93.8 | 93.87 | 92.73 | 94.23 | 94.12 |
| 1501 | 81.01 | 88.29 | 79.66 | 80.16 | 81.53 | 83.97 | 81.09 |
| 14 | 73.13 | 76.71 | 76.51 | 76.3 | 75.2 | 72.6 | 76.36 |
| 1497 | 80.19 | 90.15 | 80.88 | 80.14 | 80.18 | 82.09 | 98.96 |
| 1494 | 84.13 | 84.11 | 84.46 | 84.65 | 84.39 | 83.67 | 83.11 |
| 1489 | 81.95 | 84.81 | 83.27 | 82.72 | 82.04 | 84.18 | 83.97 |
| 1487 | 93.71 | 93.72 | 93.59 | 93.41 | 93.68 | 93.7 | 94.02 |
| 1486 | 94.75 | 94.64 | 94.75 | 94.84 | 94.83 | 94.81 | 95.38 |
| 1485 | 53.69 | 66.82 | 67.1 | 66.82 | 62.54 | 53.86 | 61.14 |
| 1480 | 70.5 | 69.57 | 68.72 | 69.42 | 69.62 | 70.63 | 68.52 |
| 1478 | 93.82 | 94.06 | 93.92 | 94.41 | 93.68 | 93.45 | 96.9 |
| 1475 | 49.24 | 50.63 | 49.67 | 49.35 | 49.65 | 49.06 | 52.08 |
| 1468 | 85.01 | 88.74 | 80.14 | 74.45 | 77.73 | 71.39 | 81.67 |
| 1464 | 75.26 | 75.37 | 75.7 | 75.36 | 75.4 | 75.17 | 75.82 |
| 1462 | 98.48 | 99.13 | 98.9 | 98.78 | 99.39 | 99.58 | 97.36 |
| 1461 | 89.87 | 89.94 | 90.04 | 89.94 | 89.92 | 89.81 | 90.22 |
| 12 | 91.83 | 94.38 | 92.78 | 92.68 | 92.95 | 92.86 | 91.49 |
| 11 | 88.38 | 87.99 | 85.48 | 85.84 | 89.13 | 84.27 | 82.43 |
| 1068 | 92.97 | 92.75 | 92.97 | 92.91 | 92.93 | 93 | 92.39 |
| 1067 | 84.88 | 84.87 | 84.77 | 84.7 | 84.27 | 84.06 | 84.24 |
| 1063 | 82.22 | 82.78 | 81.81 | 81.3 | 80.33 | 82.38 | 81.21 |
| 1053 | 80.82 | 80.91 | 80.87 | 80.89 | 80.95 | 80.94 | 80.98 |
| 1050 | 90.2 | 89.3 | 89.46 | 89.17 | 89.06 | 89.73 | 88.43 |
| 1049 | 87.59 | 87.14 | 87.33 | 87.47 | 88.11 | 87.95 | 89.18 |

Table 4: Absolute test accuracy (percent; averaged over 30 trials) for the 25% training data (semi-supervised) setting, for each dataset and every baseline, including a gradient-boosted decision tree (XGBoost) baseline (max depth of 3, 100 estimators).

| dataset id | control | SCARF | SCARF AE | add. noise AE | no noise AE | SCARF disc. |
|---|---|---|---|---|---|---|
| 6 | 20.1 | 17.23 | 19.3 | 18.42 | 18.77 | 18.1 |
| 6332 | 10.39 | 9.72 | 12.93 | 12.53 | 11.1 | 9.85 |
| 54 | 12.03 | 12.68 | 11.83 | 12 | 11.5 | 13.95 |
| 50 | 12.92 | 13.58 | 10.63 | 11.65 | 14.63 | 9.02 |
| 46 | 9.32 | 7.8 | 7.55 | 7.15 | 7.53 | 7.93 |
| 469 | 7.36 | 7.38 | 7.43 | 7.02 | 6.97 | 7.9 |
| 458 | 7.62 | 5.17 | 6.12 | 6.03 | 7.03 | 6.57 |
| 4538 | 13.17 | 15.18 | 16.3 | 15.38 | 14.97 | 14.13 |
| 4534 | 14.03 | 12.1 | 14.88 | 13.93 | 13.2 | 12.68 |
| 44 | 8.86 | 8.55 | 9.85 | 9.3 | 9.78 | 7.52 |
| 4134 | 9.13 | 12.45 | 11.07 | 11.05 | 10.85 | 10.48 |
| 41027 | 31.3 | 33.37 | 35.62 | 37.9 | 37.62 | 36.2 |
| 40994 | 5 | 5.3 | 5.37 | 6.03 | 5.28 | 5 |
| 40984 | 13.46 | 13.7 | 13.47 | 13.02 | 13.23 | 13.53 |
| 40983 | 9.91 | 7.78 | 8.8 | 9.3 | 8.45 | 6.75 |
| 40982 | 10.94 | 10.45 | 12.5 | 12.38 | 11.45 | 12.57 |
| 40979 | 10.65 | 7.77 | 10 | 10 | 10.65 | 10.32 |
| 40978 | 13 | 13.38 | 15.3 | 13.15 | 13.92 | 13.3 |
| 40975 | 15.09 | 12.63 | 14.12 | 13.78 | 14.13 | 15.33 |
| 40966 | 12.62 | 11.37 | 12.28 | 11.7 | 11.1 | 13.87 |
| 40923 | 5 | 5 | 5 | 5 | 5 | 5 |
| 40701 | 10.53 | 12.72 | 11.25 | 11.48 | 11.72 | 10.38 |
| 40670 | 9.91 | 8.93 | 10.4 | 9.92 | 9.72 | 10.27 |
| 40668 | 10.93 | 12.3 | 11.45 | 12.07 | 12.15 | 11.92 |
| 40499 | 13.19 | 10.95 | 12.55 | 12 | 11.35 | 13.17 |
| 3 | 10.72 | 12.07 | 11.9 | 11.5 | 12.5 | 11.63 |
| 38 | 9.53 | 9.03 | 9.83 | 10.53 | 9.9 | 9.88 |
| 37 | 8.11 | 7.33 | 7.15 | 7.53 | 7.77 | 7.05 |
| 32 | 10.73 | 9.73 | 10.68 | 10.35 | 11.3 | 10.62 |
| 31 | 8.45 | 8.35 | 9.58 | 9.13 | 9.87 | 7.5 |
| 307 | 17.08 | 13.27 | 16.78 | 16.7 | 17.43 | 16.88 |
| 300 | 12.2 | 16.5 | 13.38 | 14.4 | 13.28 | 14.4 |
| 29 | 6.92 | 8.05 | 8.57 | 9.48 | 9.15 | 7.38 |
| 28 | 10.68 | 9.85 | 11.83 | 12.15 | 11.6 | 10.98 |
| 23 | 9.06 | 8.05 | 10.38 | 10.48 | 10.65 | 7.8 |
| 23517 | 8.37 | 7.83 | 8.45 | 8.4 | 7.8 | 7.62 |
| 23381 | 7.88 | 6.67 | 7.9 | 6.95 | 7.17 | 6.83 |
| 22 | 10.81 | 9.37 | 11.55 | 10.48 | 10.05 | 12.2 |
| 18 | 13.2 | 10.52 | 12.73 | 11.93 | 11.07 | 12.52 |
| 188 | 10.65 | 14.62 | 12.62 | 11.6 | 11.7 | 12.15 |
| 182 | 12.23 | 12.35 | 13.4 | 12.6 | 12.03 | 14.77 |
| 16 | 12.68 | 8.5 | 10.25 | 10.63 | 11.52 | 10.82 |
| 15 | 5.59 | 5.3 | 6 | 6.05 | 7.02 | 6.72 |
| 1590 | 7.5 | 7.12 | 7.4 | 7.67 | 7.53 | 7.33 |
| 151 | 19.5 | 14.55 | 18.27 | 19.07 | 18.37 | 16.77 |
| 1510 | 7.24 | 6.67 | 7.42 | 7.37 | 7.85 | 7.25 |
| 1501 | 11.84 | 9.75 | 12.48 | 12.32 | 11.97 | 12.53 |
| 14 | 10.65 | 10.02 | 11.22 | 11.38 | 11.38 | 12.12 |
| 1497 | 13.42 | 11.82 | 15.87 | 14.83 | 14.45 | 13.85 |
| 1494 | 8.68 | 8.28 | 8.65 | 8.55 | 8.25 | 9.02 |
| 1489 | 13.84 | 12.52 | 14.12 | 13.97 | 14.62 | 12.6 |
| 1487 | 7.81 | 7.98 | 7.93 | 8.17 | 7.33 | 7.75 |
| 1486 | 11.3 | 11.35 | 11.05 | 11.6 | 10.33 | 11.5 |
| 1485 | 8.03 | 8.42 | 7.77 | 7.35 | 7.02 | 8.53 |
| 1480 | 5.44 | 6.63 | 6.75 | 6.57 | 6.6 | 6.25 |
| 1478 | 11.4 | 13.8 | 12.38 | 12.17 | 11.78 | 12.57 |
| 1475 | 13.16 | 11.57 | 14.57 | 13.65 | 13.42 | 14 |
| 1468 | 12.25 | 11.12 | 14.1 | 16.35 | 14.93 | 13.62 |
| 1464 | 7.83 | 8.18 | 8.07 | 8.6 | 6.95 | 8.12 |
| 1462 | 8.03 | 6.4 | 7.58 | 7.43 | 7.28 | 5.68 |
| 1461 | 7.53 | 6.92 | 7.75 | 7.5 | 8.02 | 6.45 |
| 12 | 11.16 | 8.13 | 9.73 | 10.07 | 9.7 | 11.68 |
| 11 | 10.31 | 10.97 | 11.48 | 10.78 | 9.58 | 11.38 |
| 1068 | 5.69 | 6.78 | 6.18 | 6.3 | 6 | 6.47 |
| 1067 | 7.97 | 9.87 | 8.23 | 8.55 | 8.1 | 9.03 |
| 1063 | 6.45 | 6.6 | 6.28 | 6.3 | 7.03 | 6.8 |
| 1053 | 8.44 | 8.78 | 8.12 | 9.07 | 8.43 | 9.1 |
| 1050 | 5.13 | 7.27 | 7.18 | 6.95 | 6.53 | 5.88 |
| 1049 | 8.68 | 10.92 | 9.38 | 9.42 | 8.8 | 9.5 |

Table 5: Number of actual training epochs used (averaged over 30 trials) for the 100% training data setting, for each dataset and every baseline.

