# OpenReview forum: "Scarf: Self-Supervised Contrastive Learning using Random Feature Corruption"
_ICLR.cc/2022/Conference — ICLR 2022 Spotlight_

### Official Review · Reviewer_6ZpH · 2021-10-17

**Correctness:** 4
**Technical Novelty And Significance:** 2
**Empirical Novelty And Significance:** 4
**Recommendation:** 8
**Confidence:** 4

**Main Review:**

There are several merits of the paper:
 - While the augmentation scheme looks very simple and intuitive, it is superior to all conceivable baselines.
 - Thorough evaluation scheme using 'win matrix', measured how many datasets SCARF beats other baselines. To compute the average performance of the algorithm, they run 30 times on each dataset and method pair
- Wide range of findings - showed its superiority in label-corrupted and partially-labeled settings, compared between several corruption strategies, done sensitivity analysis with regards to key hyperparameters.
 - As for the experiment, the authors also explored recently-proposed loss alternatives including Uniform and Align and Barlow Twins, and showed that InfoNCE is a good choice.

The paper could be more complete if there was a discussion between two self-supervised learning approaches for tabular data: contrastive learning and pre-text learning models such as VIME (NeurIPS 2020) or TabNET (AAAI 2021).

**Summary Of The Paper:**

This paper proposes SCARF, which is a contrastive pretraining procedure for tabular data. SCARF generates an augmentation of a data point by selecting a random subset of features and replace them with their marginal distributions. Unsupervised pre-training is done by update networks consists of an encoder and pretraining head. Contrastive InfoNCE loss is used to pull the original input and its perturbed version, while pushing it away from other data points in mini-batch. On 69 datasets from OpenML-CC18 benchmarks, SCARF is compared with dozens of baselines and showed its efficacy. SCARF pretraining improves model robustness in label-corrupted settings and also helps classification in partially-labeled settings.

**Summary Of The Review:**

Simple and intuitive contrastive learning framework for tabular data, backed up by extensive experiments. Considering the sufficient experiments, the proposed SCARF algorithm is expected to perform well on any tabular dataset, and the simplicity of the algorithm is expected to draw attention and will bring subsequent methodologies.

---

> ### Author Response · Authors · 2021-11-23
> **Author Response to Reviewer 6ZpH**
>
> We thank you for the thoughtful review!
> Both VIME and TabNet are very relevant to our paper and we will update the related works to reflect this for the camera-ready.

---

> > ### Comment · Reviewer_6ZpH · 2021-12-01
> > **Response to Authors**
> >
> > Are you planning to release the codes for SCARF and experiments? It would be helpful for communities.

---

> > > ### Author Response · Authors · 2021-12-01
> > > **RE: releasing code**
> > >
> > > Thanks for the suggestion, we will release the code upon legal clearance.

---

### Official Review · Reviewer_y3t1 · 2021-10-18

**Correctness:** 3
**Technical Novelty And Significance:** 3
**Empirical Novelty And Significance:** 3
**Recommendation:** 8
**Confidence:** 4

**Main Review:**

Strengths:

(1) The motivation of this paper is clear and the problem they studied is valuable. InfoNCE loss (or the contrastive learning paradigm) has been well studied in computer vision but it is not obvious to design augmentations (i.e., generate different views) for tabular data. Images are invariant under various transformations (e.g., translation, rotation, color jittering or cropping) but it is hard to design such transformations for tabular data.

(2) The experimental results are very sufficient and the authors provide a wide range of comparison methods. They experimented with 69 datasets and repeated 30 times for each run. Hence, I think the experimental results are convincing.

(3) The authors investigate the label-noise and semi-supervised setting. There has been works in computer vision that have studied applying SSL under label-noise and semi-supervised setting, but this is the first time I have seen for the tabular datasets.

(4) Some conclusions given by the authors are instructive for further study. For instance, corrupting one view is better than corrupting both the views for tabular.

-------------------------------------------------------------------------------------------------------------------------------
Weakness:

(1) When comparing accuracies with and without self-supervised pre-training, is the number of fine-tuning epochs the same?  As noted in the paper, the authors set a max number of fine-tune epochs of 200 and pre-train epochs of 1000. Should we fine-tune for longer epochs (i.e., >200) if we don't conduct self-supervied pre-training for fair comparisons?

(2) Is label information useful when sampling? For instance, when using joint sampling or marginal samling, what if we consider the label $y_i$ for $x_i$ and only consider the instances in the same class when sampling? Although pre-training is not unsupervised in this case, I don’t think this is a problem because labels are also used in the fine-tuning phase.

(3) I was curious about whether SCARF pre-training outperforms using SCARF augmentation during fine-tuning and I appericiate the authors for providing such results. However, as noted in weakness (1), the training details (especially epochs) are not clear and I am not sure whether it is a fair comparison.

(4) Some conclusions need further explainations. For instance, it is common practice in computer vision to corrupt both views during SSL but why corrupting one view is better than corrupting both views in this paper. Is it due to the characteristics of the tabular data or the proposed method?


**Summary Of The Paper:**

This paper proposes a data augmentation method (SCARF) used for self-supervised learning for tabular data. SCARF generates different views by corrupting a random subset of features (via sampling from the marginal distribution). The experimetanl results demonstrate that SCARF not only improves accuracy in the fully-supervised setting but also in the presence of label noise and in the semi-supervised setting. The authors also conduct ablation studies and compare with various methods.

I think adapting SSL to tabular data is a very important direction and I thank the authors for their efforts in this direction.

**Summary Of The Review:**

This paper is well-writtern and the motivation of paper is clear. I agree with the contributions of this paper but I still have some concerns (see the weakness). I am happy to raise my score if my concerns are addressed.

---

> ### Author Response · Authors · 2021-11-23
> **Author Response to Reviewer y3t1**
>
> Thank you for your thorough review.
>
> **RE: fine-tuning epochs**
>
> This is a good point. Recall that we use an early stopping criteria for both (SCARF) pre-trained and non-pre-trained models. The number of epochs needed is far smaller than the 200 limit. To show this, **we have updated the Appendix with a full-page table showing the number of fine-tuning epochs (averaged over the 30 trials) required (per early stopping) for each dataset for the 100% training data setting. We see that SCARF pre-trained models often need fewer fine-tuning epochs than control.** We believe that comparisons wherein training steps are artificially limited so as to put the proposed method and alternatives on equal computational footing can be fraught with error. Instead, we ask the practical question: given a task and a reasonable choice of model (that neither underfits or overfits), and generous computational resources, what's the best you can do?
>
> **RE: "Is label information useful when sampling?"**
>
> This is a great suggestion! We did not explore leveraging label information during pre-training as a desideratum we had in mind from the beginning is that the method should be unsupervised (so that it can be leveraged when you have unlabeled data available or when labels are corrupted). We leave supervised pre-training for future work.
>
> **RE: SCARF Pre-training vs. augmentation.**
>
> We are glad you found this comparison insightful. We hope the aforementioned table (in the Appendix) showing training epochs until convergence may help address your concern here.
>
> **RE: Clarifying why corrupting one view is better than corrupting both views in this paper.**
>
> It's challenging to pinpoint exactly why corrupting one view is better than both, but we suspect the corruption method and the data-generating distribution / data dimensionality are at play. Practically speaking, corrupting two views with SCARF can be similar to corrupting one view but at different corruption rates. Let $N$ be the number of features. Then with one-view-corruption with corruption rate $c$, the expected number of shared feature entries between views is $N*(1-c)$. Now, with two-view-corruption each with rate $k$, this quantity is $N*(1-k)^2$. So a similar effect may be achieved by equating these two quantities, i.e. $k = 1 - \sqrt{1-c}$.
> When developing SCARF, we often worked backwards to simplify the method as much as possible -- that is, to drop complexities for which experiments do not show meaningful gains.

---

> > ### Comment · Reviewer_y3t1 · 2021-11-23
> > **Response to Authors**
> >
> > Thank you for your response. They go a long way in addressing my concerns about the paper so I have amended my rating to reflect the response.

---

### Official Review · Reviewer_7Frm · 2021-10-24

**Correctness:** 3
**Technical Novelty And Significance:** 3
**Empirical Novelty And Significance:** 3
**Recommendation:** 6
**Confidence:** 4

**Main Review:**


Strengths:
1) Easy to read paper with very comprehensive results across a variety of datasets. They also try >10 baselines and show very comprehensive improvements on top of these baseline methods.
2) SCARF also shows improved performance even in presence of label noise, which suggests that SCARF is also more robust to corruption in the datasets as well.
3) SCARF is also less sensitive to feature scaling as compared to previous methods.

Weakness:
1) Role of negative samples: one thing I’m not sure about is what is the role of negative samples in SCARF. From the results in Figure 5, it can be seen that increasing the batch size doesn’t really have an impact on the performance. Can it be that a negative sample doesn't have much impact on the results? It would be interesting to see a ablation showing the performance dip as we decrease the #negative samples.
2) Results using other non contrastive methods:  Results using BYOL and other non-contrastive methods would be interesting to see. I can see that Barlow twins perform very similarly to Info-NCE methods, but I’m not sure how many hyperparameter searches have been made on Barlow-Twins. Generally I’m asking if it is possible to outperform infonce methods with non contrastive methods. Even if it doesn't work well; it would be nice to see some analysis on why non-contrastive methods don’t work as well on tabular data while they are working better on Image datasets.
3) Temperature = 1: usually in contrastive learning temperature plays a huge role in representation ability of the network. But in this case the temp is 1, which effectively means there is no effect of temperature. Can the authors comment more on that ?


**Summary Of The Paper:**

This work proposes a good method to use SSL methods on tabular datasets. They achieve this by randomly corrupting a subset of the features. They show comprehensive results and improved performances on 69 real-world, tabular classification datasets from the OpenML-CC18 benchmark.


**Summary Of The Review:**

Overall I like the paper, there are few concerns regarding use of negative samples and below par performance on non contrastive methods. It would be nice to have some more analysis on this front.

---

> ### Author Response · Authors · 2021-11-23
> **Author Response to Reviewer 7Frm**
>
> Thank you for the review! We are glad you enjoyed the paper.
>
> **RE: "Can it be that a negative sample doesn't have much impact on the results? It would be interesting to see a ablation showing the performance dip as we decrease the #negative samples."**
>
> The ablation on batch size (Figure 5) is in fact an ablation on the # negative samples, since a batch size of N means that the contrastive loss term will have 1 positive to N-1 negatives (as mentioned in the paragraph titled "SCARF is not sensitive to batch size"). As described in the paper, on the tasks and model sizes we explore in this work, using more than 128-1 negatives for every positive did not result in significant improvements, but going too small hurts (for example 128 batch size beats batch size 4 37 out of 37 times and batch size 16 21 out of 21 times). Given that SimCLR showed benefits of large-batch / large-negative samples for ResNet50 on ImageNet, as discussed in our paper, a reasonable hypothesis is that higher capacity models and harder tasks benefit more from negatives.
>
> **RE: "I can see that Barlow twins perform very similarly to Info-NCE methods, but I’m not sure how many hyperparameter searches have been made on Barlow-Twins."**
>
> For uniform-align, we tried (a, u) = (0, 1), (1, 0), (1, 1), (1, 2), (1, 3), where a is the weight on the align term and u is the weight on the uniform term.
> For Barlow, we used the original paper's best reported hyper-parameter $\lambda$=5e-3.
>
> **RE: "Results using BYOL and other non-contrastive methods would be interesting to see"**
>
> This is a great suggestion, thank you! Implementing BYOL in our experimental setup is bit more involved than we have time for during the rebuttal period, but this is a comparison we could potentially do for the camera-ready if the paper is accepted.
>
> **RE: temperature = 1 is optimal.**
>
> A temperature T is used to scale the similarities for the positive and negative pairs before taking a softmax. Large T has the effect of attenuating the loss when the positive pair similarity is larger than the negative pair similarity. Whether to use temperature and what setting to use is intricately tried to the factors like batch size / number of negatives, training epochs (the SimCLR paper has plots for this), model architecture, dataset, as well as (crucially) the similarity measure (which depends on the corruption strategy), so it's hard to pinpoint precisely why we don't need it in the settings we explore in our paper. With that said, we find it a blessing to the practitioner that this hyper-parameter does not need much (or any) tuning in our setup.

---

### Official Review · Reviewer_swwj · 2021-11-02

**Correctness:** 4
**Technical Novelty And Significance:** 2
**Empirical Novelty And Significance:** 3
**Recommendation:** 6
**Confidence:** 4

**Main Review:**

Strengths:
- Exploring self-supervision on novel domains (e.g., tabular data) is valuable
- The method for generating ‘augmentations’ for contrastive learning via random replacement from the marginal feature distribution makes sense and is novel to the best of my knowledge
- The idea is simple but seems quite effective
- The paper is well written, and the method is represented well
- Experiments show relative improvements over several baselines

Weaknesses:
- All the experiments only show relative comparisons. It is also important to get a sense of the absolute performance on these benchmarks, e.g., to assess how strong the baseline is compared to the existing state-of-the-art. An obvious baseline on tabular data would be a tree ensemble model (like XGBoost).
- Some of these relative comparisons are also difficult to interpret: It is, for example, unclear what the second Win-matrix is showing. Is it SCARF pre-training vs. Dropout or SCARF with vs. without Dropout? How does it follow that SCARF is complementary to Dropout from the Figure when we do not see any absolute performance numbers?
- The method from Yao et al. (2020) sounds quite similar, but it is not clear what the advantage of SCARF would be, and a comparison is also missing.
- The joint sampling ablation is flawed if the replacement rate is set to the default c=0.6. This would only make sense when c<<0.5. The augmented sample otherwise has more resemblance with the randomly drawn instance.

**Summary Of The Paper:**

The paper presents a method for self-supervised pre-training on tabular data to improve the performance in supervised transfer.
The method adapts the successful contrastive learning framework to tabular data by defining “augmentations” of the data wherein randomly chosen feature columns are replaced by sampling from their corresponding marginal distribution.
The method is evaluated on tabular classification tasks of the OpenML-CC18 benchmark, both in the fully supervised setting and in the semi-supervised setting and under 30% label noise.
The method outperforms several baselines.

**Summary Of The Review:**

The paper introduces a simple adaptation of contrastive learning to tabular data by introducing a novel ‘data augmentation’ on tabular data. While the method outperforms several baselines, the experimental setup is not very convincing. Strong baselines like XGBoost are missing, a comparison to the related prior work by Yao et al. is missing, and only relative performance improvements over baselines are reported while absolute performance numbers are missing altogether.

---

> ### Author Response · Authors · 2021-11-23
> **Author Response to Reviewer swwj**
>
> Thank you for your thoughtful review; we are pleased to hear you find this work novel, valuable, and well-written.
>
> **RE: absolute performance and gradient boosted decision tree (xgboost) baseline.**
>
> This is a good point. The reason why we decided to present relative gains instead of absolute performance is that it's hard to summarize absolute performance across the 69 datasets we consider, whereas relative gains are not. However, we understand how absolute numbers might be needed for a complete picture, so **we have updated the Appendix with 3 full-page tables that show the absolute test accuracy (averaged over the 30 trials) for each of the 69 datasets for SCARF and baseline methods for each of the 3 settings we consider (100% data, 30% label noise, 25% training data). Furthermore, we follow your suggestion and add an XGBoost baseline and show its absolute performance**. SCARF-enabled deep nets often outperform XGBoost, sometimes when deep nets without SCARF cannot. For example, on dataset 50, control (vanilla deep net) achieves 90.54%, xgboost gets 90.61%, but deep nets pre-trained with SCARF achieve an eye-popping 97.53%. We hope these tables provide all the details you were looking for. We used XGBoost's Python API (https://xgboost.readthedocs.io/en/latest/python/python_api.html#xgboost.XGBClassifier), version 0.6, with the default settings: max_depth=3, n_estimators=100, learning_rate=0.1.
>
> **RE: "Is it SCARF pre-training vs. Dropout or SCARF with vs. without Dropout?"**
>
> It is **SCARF with dropout vs. dropout**. In the second win matrix, rows are SCARF + baseline approaches, whereas columns are common, well-established techniques that improve performance in the settings we consider (and as you may have noticed, we include specific techniques for specific settings, like deep knn for label noise and self-training for semi-supervised learning).
>
> **RE: "How does it follow that SCARF is complementary to Dropout from the Figure when we do not see any absolute performance numbers?"**
>
> In light of the previous point, the second win matrix shows that SCARF with dropout outperforms dropout, thus SCARF and dropout are complementary -- i.e. the two are not in competition with each other. We will spell this out more in the main text.
>
>
> **RE: "The method from Yao et al. (2020) sounds quite similar, but it is not clear what the advantage of SCARF would be, and a comparison is also missing."**
>
> The method from Yao et al. (2020), as described in their paper, is suited specifically for large-scale item recommender systems. Their method, Correlated Feature Masking (CFM), only works for categorical features, which prevents us from applying it directly to our tasks, since the feature vectors for our tasks are heterogeneous in nature -- they contain both numerical and categorical features. Other differences include but are not limited to: 1) Our method is a pre-training method whereas CFM is used for co-training (two term loss).
> 2) In our method, we corrupt by sampling from the marginal distribution whereas CFM drops them out entirely.
> While a direct comparison is not possible, we would like to say that our extensive set of baselines do have a lot of overlap with CFM. For example, we specifically test co-training vs. pre-training and find that pre-training is better. In a similar vein, we show that marginal sampling is a better corruption strategy than dropout.
>
>
> **RE: "The joint sampling ablation is flawed if the replacement rate is set to the default c=0.6. This would only make sense when c<<0.5. The augmented sample otherwise has more resemblance with the randomly drawn instance."**
>
> **This claim is not true.** Let $x$ be the original example and let $\tilde{x}$ be its corrupted view formed by sampling $z$ uniformly at random from the training dataset and then setting a random 60% of $\tilde{x}$'s entries to $z$'s entries and the remaining 40% to $x$'s entries (i.e. joint sampling with $c = 0.6$). It is true that $\tilde{x}$ will have more features in common with $z$ than with $x$, but this doesn't affect our loss since our loss is concerned with $\text{sim}(x, \tilde{x})$ and $\text{sim}(x, z)$, not $\text{sim}(\tilde{x}, z)$.

---

> > ### Comment · Reviewer_swwj · 2021-11-23
> > **Response to Authors**
> >
> > Thank you for your thorough response and the updated results! They are much appreciated.
> >
> > R1:
> > I appreciate the added absolute performance numbers. I think it might be a good idea also to summarize those (maybe via the average performance over all datasets). It is a bit difficult to get a clear picture as is. XGBoost provides quite a strong baseline even without tuning it to the tasks. While there are datasets where SCARF performs better, there are also examples where only XGBoost works (see 40923, where all the deep learning variants fail).
> >
> > R2:
> > Maybe I misunderstand your answer, but your explanation indicates that it is actually "SCARF with dropout vs. dropout" in the second win matrix and not "SCARF with dropout vs. SCARF without dropout"? This is still not entirely clear in the figures, unfortunately. The description of Table 1 is more precise.
> >
> > R3:
> > If it is "SCARF with dropout vs. dropout," as I understood it, and all the baselines indeed improve over the control (not shown as far as I can tell), then this makes sense.
> >
> > R4:
> > Thanks for pointing out the differences.
> >
> > R5:
> > I wrongly assumed that z would be sampled from the same batch and not from the whole dataset. It would only pose a problem when z is also a sample in the same batch, which should be unlikely if the batch size is much smaller than the dataset size.
> >
> > All in all, the response addresses some of my concerns. A well-tuned XGBoost baseline may still outperform most deep learning variants. It would be good to summarize absolute performance on the benchmark also in the main paper and address the remaining unclarities. I adjust my rating accordingly.

---

> > > ### Author Response · Authors · 2021-11-23
> > > **RE: Response to Authors**
> > >
> > > Thank you for reading the author response and updating your score.
> > >
> > > **RE: R1**
> > > Summarizing by averaging accuracy over all datasets may not be very meaningful because more weight will be put on high-accuracy tasks than low ones, but the number of datasets for which SCARF outperforms control is...but this is already captured in the win ratio.
> > > The figures we already show in the main text -- the win ratio and the box plots are nice in that they already capture **how often** and **by what degree** SCARF outperforms the baselines.
> > > **RE: "there are also examples where only XGBoost works"**
> > > Of course, GBDT / XGBoost is an industry standard for working with tabular data. But the claim that XGBoost is untuned whereas the deep nets we train in this work are is not true. Our deep net architecture is a vanilla ReLU feed-forward; nothing about it was tuned for the openML datasets. Furthermore, note that XGBoost, as far as I can tell, may employ regularization and heuristics (to adapt to the dataset) during training, whereas our deep nets do not. With that said though, the message to practitioners is: don't necessary abandon your trees for deep nets (you should try both for your problem), but if you do try deep nets, try pre-training with SCARF as it'll probably help.
> > >
> > > **RE: R2**
> > > We are so sorry, this was a typo in our author response (which we will edit). As you sensed, and as described in the caption for Table 1, it is in fact **SCARF with dropout vs. dropout** and not **SCARF with dropout vs. SCARF**.

---

> > > > ### Comment · Reviewer_6ZpH · 2021-11-30
> > > > **Voted for measuring the win ratio**
> > > >
> > > > 'Summarizing by averaging accuracy over all datasets may not be very meaningful because more weight will be put on high-accuracy tasks than low ones, but the number of datasets for which SCARF outperforms control is...but this is already captured in the win ratio'. -- No more agree on this part (reviewer 6Zph)

---

### Public Comment · ~Yoontae_Hwang1 · 2022-09-07
**Dataset setting**

I have some questions on the experiment setting.

I have any questions as follows :
How is the dataset organized in the pre-training and the supervision stages? From what I understand, it is as follows.
If you have a dataset id 6, configure the dataset for 20% Labeled training as follows: Full dataset = train set(for pre-training) + train set(for supervised) + validation set(for pre-training) + validation set(for supervised) + test set
As far as I understood, the dataset for pre-training and the supervision are independent with each other, so with no overlap. Am I correct?  In addition, I understood that the training set (for pre-training) + validation set (for pre-training) uses 80% of the "full dataset", and the remaining 20% is used for the supervision task. Is this correct?
 I would like to know the detail of how the XGBoost is utilized in the experiment. How did you create a 20% and 100% label configured dataset to use XGboost?

---

> ### Public Comment · ~Dara_Bahri1 · 2022-09-22
> **RE: dataset setting**
>
> There is overlap between pre-training and supervision data.
> Let x be the whole dataset and let x[a:b] mean the data with indices between a% of x.size and b% of x.size -- i.e. x[:20] means the first 20% of the dataset.
> We ran 30 trials. For each trial, we set a random seed and shuffle the entire dataset using that random seed. Below, znorm is the data used to compute z-score normalization. Once computed, it is applied to all splits including the test.
>
> x = shuffle(x, random_seed)
> train = x[:70]
> val = x[70:80]
> znorm = x[:80]
> test = x[80:]
> pretrain_train = x[:70] (no labels)
> pretrain_val = x[70:80] (no labels)
>
> For sample efficiency / semi-supervised experiments, where the training data is sub-sampled:
> x = shuffle(x, random_seed)
> s = 17.5
> train = x[:s]
> val = x[70:80]
> unlabeled_train = x[s:70]
> znorm = x[:80]
> test = x[80:]
> pretrain_train = x[:70]
> pretrain_val = x[70:80]
>
> RE: XGBoost --
> import xgboost
> model = xgboost.XGBClassifier(
>       max_depth=3,
>       objective='multi:softmax',
>       n_estimators=100,
>       num_class=num_classes)
> model.predict_proba(data)

---

### Public Comment · ~Tianping_Zhang1 · 2023-03-07
**Code release**

Dear Scarf authors,

I hope this message finds you well. First of all, I would like to express my sincere gratitude for your intriguing work. I have been trying to reproduce the results using Scarf. However, despite my best efforts to align with the experimental settings described in the paper, I have not been able to achieve the same results reported in the paper, and Scarf does not seem to be as effective as expected.

I understand that during the rebuttal, you mentioned that you would release the code upon legal clearance. I was wondering if it would be possible for you to share the codes? I would be extremely grateful for any help you could provide.

Thank you very much for your time and consideration.

---

### Decision · Program_Chairs · 2022-01-20

**Decision:**

Accept (Spotlight)

**Comment:**

The paper explores self-supervised learning on tabular data and proposes a novel augmentation method via corrupting a random subset of features. The idea is simple but effective. Experiments include 69 datasets and compare with a number of methods. The result shows its superiority. It would be inspiring more work for SSL on the tabular domain.